

# A comprehensive in situ and remote sensing data set from the Arctic CLoud Observations Using airborne measurements during polar Day (ACLOUD) campaign

André Ehrlich[1], Manfred Wendisch[1], Christof Lüpkes[2], Matthias Buschmann[3], Heiko Bozem[4],
Dmitri Chechin[2], Hans-Christian Clemen[5], Régis Dupuy[6], Olliver Eppers[4,5], Jörg Hartmann[2],
Andreas Herber[2], Evelyn Jäkel[1], Emma Järvinen[7], Olivier Jourdan[6], Udo Kästner[8],
Leif-Leonard Kliesch[9], Franziska Köllner[6], Mario Mech[9], Stephan Mertes[8], Roland Neuber[2],
Elena Ruiz-Donoso[1], Martin Schnaiter[10], Johannes Schneider[6], Johannes Stapf[1], and Marco Zanatta[2]

[1]Leipziger Institut für Meteorologie (LIM), Universität Leipzig, Leipzig, Germany
[2]Alfred–Wegener–Institut, Helmholtz–Zentrum für Polar– und Meeresforschung (AWI), Germany
[3]Institut für Umweltphysik (IUP), Universität Bremen, Bremen, Germany
[4]Institut für Physik der Atmosphäre (IPA), Johannes Gutenberg-Universität, Mainz, Germany
[5]Particle Chemistry Department, Max-Planck-Institut für Chemie (MPIC), Mainz, Germany
[6]Laboratoire de Météorologie Physique (LaMP), Université Clermont Auvergne/ OPGC/CNRS, UMR 6016,
Clermont-Ferrand, France
[7]National Center for Atmospheric Research (NCAR), Boulder, CO, USA
[8]Leibniz–Institut für Troposphärenforschung (TROPOS), Leipzig, Germany
[9]Institut für Geophysik und Meteorologie (IGM), Universität zu Köln, Cologne, Germany
[10]Institut für Meteorologie und Klimaforschung, Karlsruher Institut für Technologie (KIT), Karlsruhe, Germany

**Correspondence:** André Ehrlich (a.ehrlich@uni-leipzig.de)

**Abstract.**

The Arctic Cloud Observations Using Airborne Measurements during Polar Day (ACLOUD) campaign was carried out North-West of Svalbard (Norway) between 23 May - 26 June 2017. The objective of ACLOUD was to study Arctic boundary layer and mid-level clouds and their role in Arctic Amplification. Two research aircraft (Polar 5 and 6) jointly performed 22 research flights over the transition zone between open ocean and closed sea ice. Both aircraft were equipped with identical instrumentation for measurements of basic meteorological parameters, as well as for turbulent and and radiative energy fluxes. In addition, on Polar 5 active and passive remote sensing instruments were installed, while Polar 6 operated in situ instruments to characterize cloud and aerosol particles as well as trace gases. A detailed overview of the specifications, data processing, and data quality is provided here. It is shown, that the scientific analysis of the ACLOUD data benefits from the coordinated operation of both aircraft. By combining the cloud remote sensing techniques operated on Polar 5, the synergy of multi-instrument cloud retrieval is illustrated. The remote sensing methods are validated using truly collocated in situ and remote sensing observations. The data of identical instruments operated on both aircraft are merged to extend the spatial coverage of mean atmospheric quantities and turbulent and radiative flux measurement. Therefore, the data set of the ACLOUD campaign provides comprehensive in situ and remote sensing observations characterizing the cloudy Arctic atmosphere. All processed,





calibrated, and validated data are published in the world data center PANGAEA as instrument-separated data subsets (Ehrlich et al., 2019b, https://doi.org/10.1594/PANGAEA.902603).

## 1   Introduction

The considerable increase of Arctic near-surface temperatures within the last three to four decades, a phenomenon commonly called Arctic amplification (Serreze and Barry, 2011), significantly exceeds the global warming and is associated with the decrease of Arctic sea ice. To improve the understanding and the abilities to predict these changes, several international efforts including joint model evaluations such as the Year of Polar Prediction within the Polar Prediction Project (Jung et al., 2016) and a series of observational field campaigns are underway. These observations obtained by land-based (Uttal et al., 2016),

ship-based, and airborne activities (Wendisch et al., 2019) are essential to identify the dominant atmospheric processes and provide an observational basis for model and satellite data validations. Due to the diversity of instrumentation and required measurement strategies, these field campaigns often target specific components of the Arctic climate system.

In May/June 2017 two concerted field studies, the Arctic Cloud Observations Using Airborne Measurements during Polar Day (ACLOUD) campaign and the Physical Feedbacks of Arctic Boundary Layer, Sea Ice, Cloud and Aerosol (PASCAL) ship

cruise have been performed to improve our understanding of the role of clouds and aerosol particles in Arctic amplification (Wendisch et al., 2019). Both campaigns were conducted within the framework of the "Arctic Amplification: Climate Relevant Atmospheric and Surface Processes, and Feedback Mechanisms (AC)[3]" project (Wendisch et al., 2017). During ACLOUD, two research aircraft, Polar 5 and Polar 6 (Wesche et al., 2016) were operated, which were stationed on Svalbard (Longyearbyen, Norway). For PASCAL the Research Vessel (R/V) Polarstern (Knust, 2017) entered the sea ice north of Svalbard where an

ice floe camp (including a tethered balloon, ground-based remote sensing, and in situ sampling of aerosol particles) was setup for two weeks (Macke and Flores, 2018). These observations were accompanied by permanent measurements at the joint research base AWIPEV at Ny-Ålesund/Svalbard (Neuber, 2006) operated by Alfred Wegener Institute (AWI) and the French Polar Institute Paul-Émile Victor (IPEV; AWIPEV). The airborne operations during ACLOUD where coordinated with the ship (PASCAL) and ground-based activities (AWIPEV) and focused on the area north-west of Svalbard linking the observations at

AWIPEV and on Polarstern.

The general objectives of ACLOUD/PASCAL, the operated instrumentation, a summary of the measurement activities, and first highlights of the data analysis are presented by Wendisch et al. (2019) while the meteorological conditions during the observational period have been analyzed by Knudsen et al. (2018). In this paper, a detailed overview on the processed ACLOUD data set obtained on board of both research aircraft is provided. The aim is to document the campaign-specific instrument

operation, data processing, uncertainties of the derived quantities, and data availability to facilitate a widespread use of the data in a broad field of scientific analysis. The instrumentation, calibration, and data processing of measurements on Polar 5 and 6





are described in Sections 2 and 3. Due to the operation of two identical aircraft (partly with identical instrumentation), several benefits arise for the data analysis. Coordinated observations from both aircraft flying in close collocation, e.g., remote sensing and in situ measurements, have been combined as demonstrated in Section 4.1. In Section 4.2, the comparability of data from similar instruments operated on both aircraft is validated, which allows merging observations from both aircraft into a single

5     data set. The data availability including links to the published data sets is given in Section 5.

**Table 1.** Overview of ACLOUD flights including the takeoff and landing times of Polar 5 and 6 and the general scientific target of the flight. The columns indicate if Polar 5 and 6 overflew Polarstern, flew in collocated formation (Polar 5 above Polar 6), performed vertical profiling by staples flight patterns, or were coordinated with an A-Train overpass.

| # | Date in 2017 | Takeoff–Landing (UTC) Polar 5 | Polar 6 | Scientific Target | Collocated | Staples | Polarstern | A-Train |
|---|---|---|---|---|---|---|---|---|
| 4 | 23 May | 09:12–14:25 | – | Clouds above sea ice and open ocean | | | | |
| 5 | 25 May | 08:18–12:46 | – | Remote sensing of different cloud regimes | | | | |
| 6 | 27 May | 07:58–11:26 | – | Clouds over sea ice and open ocean | | | | X |
| 7 | 27 May | 13:05–16:23 | 13:02–16:27 | Clouds over sea ice and open ocean | X | | | |
| 8 | 29 May | 04:54–07:51 | 05:11–09:17 | Thin low level clouds over sea ice | | | | |
| 9 | 30 May | – | 09:18–13:30 | Aerosol column and mapping | | | X | |
| 10 | 31 May | 15:05–18:57 | 14:59–19:03 | Thin low-level clouds over sea ice | | | X | |
| 11 | 02 June | 08:13–13:55 | 08:27–14:09 | Low clouds in warm air over sea ice | X | | X | X |
| 12 | 04 June | – | 10:06–15:39 | Low clouds in warm air over sea ice | | | X | X |
| 13 | 05 June | 10:48–14:59 | 10:43–14:44 | Low clouds in warm air over sea ice | X | | X | |
| 14 | 08 June | 07:36–12:51 | 07:30–13:20 | Thin broken clouds over sea ice | X | | X | X |
| 15 | 09 June | 08:00–09:21 | 07:56–09:18 | P5/P6 Instrument comparison | X | | | |
| 16 | 13 June | 14:56–16:55 | 14:57–17:16 | P5/P6 Calibration | X | | | |
| 17 | 14 June | 12:48–18:50 | 12:54–17:37 | Boundary layer profiling | X | X | X | |
| 18 | 16 June | 04:45–10:01 | 04:40–10:31 | Low clouds over sea ice | | | X | X |
| 19 | 17 June | 09:55–15:25 | 10:10–15:55 | Clouds above sea ice and open ocean | X | | | |
| 20 | 18 June | 12:03–17:55 | 12:25–17:50 | Clouds above sea ice and open ocean | X | | X | |
| 21 | 20 June | 07:30–13:55 | 07:37–13:27 | Boundary layer profiling | | X | X | |
| 22 | 23 June | 10:57–14:39 | 10:37–14:52 | Column over Ny-Ålesund | X | | | |
| 23 | 25 June | 11:09–17:11 | 11:03–16:56 | Boundary layer profiling cloud-free | | X | | |
| 24 | 26 June | – | 08:33–10:39 | P6 Calibration | | | | |
| 25 | 26 June | 12:34–15:17 | 12:32–14:48 | Boundary layer profiling | X | | | |

The ACLOUD aircraft campaign performed 22 research flights between 23 May - 26 June 2017 which are listed in Table 1 (flight numbers start with #4 neglecting the test and ferry flights #1-3). In total, 165 flight hours were spent by both aircraft. A joint operation of Polar 5 and 6 was coordinated for 16 research flights. Eleven closely collocated flights with Polar 5



performing remote sensing in high altitudes (up to 4 000 m) and Polar 6 sampling clouds below (down to 70 m above sea level) were conducted (column "collocated" in Table 1). During three flights, profiles of turbulent and radiative fluxes in the atmospheric boundary layer were measured. Therefore, both aircraft flew similar vertical patterns (staples with horizontal flight legs) when horizontally separated from each other by 20-50 km (column "staples"). The flight activities were coordinated with

5 the PASCAL campaign of the research vessel Polarstern which was met ten times (column "Polarstern"). Five research flights contain legs orientated parallel to and time synchronized with overpasses of the A-Train satellite constellation (Stephens et al., 2018, column "A-Train").

## 2 Instrumentation on Polar 5

A comprehensive overview of airborne instrumentation in general is given by Wendisch and Brenguier (2013). Numerous

of the instruments installed on Polar 5 and 6 are described in detail in this reference. Polar 5 was primarily operated as a remote sensing aircraft. Active radar and lidar observations were combined with passive spectral solar and microwave sensors including an imaging spectrometer, a fish-eye camera, a microwave radiometer, and a Sun-photometer. For measurements of turbulent and radiative energy flux densities, a nose boom, and broadband solar and terrestrial radiation sensors (pyranometer and pyrgeometer) were installed. Profiles of meteorological parameters were collected by dropsondes. The instrumentation is

listed in Table 2.

Table 2: Overview of the instrumentation of Polar 5 and 6 and the measured quantities that are part of the data base. $\lambda$ is wavelength, $\nu$ is frequency, $T$ is temperature, and $p$ is atmospheric pressure. RH is relative humidity, FOV is field of view, PNSD is the particle number size distribution, rBC refractory black, and $D_\mathrm{p}$ symbolize the particle diameter

| Aircraft | Instrument | Measured quantities, range, and sampling frequency |
| --- | --- | --- |
| **Meteorology** | | |
| P5 | Dropsondes (RS904) | Profiles of $T$, $p$, RH, Horizontal Wind Vector, 1 Hz |
| **Turbulence** | | |
| P5&P6 | Nose-Boom Sensors | $T$, $p$, Wind Vector, 100 Hz |
| **Radiation** | | |
| P5&P6 | CMP-22 Pyranometer | Solar Irradiance (Upward, Downward, Broadband $\lambda = 0.2 - 3.6\,\mu m$), 20 Hz |
| P5&P6 | CGR-4 Pyrgeometer | Terrestrial Irradiance (Upward, Downward, Broadband $\lambda = 4.5 - 42.0\,\mu m$), 20 Hz |
| P5&P6 | KT-19 | Brightness Temperature (Upward nadir, $\lambda = 9.6 - 11.5\,\mu m$), 20 Hz |
| **Remote Sensing** | | |
| P5 | SMART-Albedometer | Spectral Irradiance (Upward, Downward $\lambda = 400 - 2155\,nm$), 2 Hz |
| | | Spectral Radiance (Upward, FOV $= 2.1°$, $\lambda = 400 - 2155\,nm$), 2 Hz |
| P5 | AISA Eagle/Hawk | Spectral Radiance (Upward, Swath $= 36°$, $\lambda = 400 - 2500\,nm$), 20-30 Hz |
| P5 | 180° Fish-Eye Camera | Spectral Radiance (Lower Hemisphere, RGB Channels), 6 s |
| P5 | AMALi | Particle Backscattering Coefficient ($\lambda = 355, 532\,nm$), Cloud Top Height, |





| | | |
|---|---|---|
| | | Particle Depolarization ($\lambda = 532\,\mathrm{nm}$), 1 s |
| P5 | MiRAC-A | Radar Reflectivity Factor, Doppler Spectra, $\nu = 94\,\mathrm{GHz}$, tilted by $25°$, 1-2 s |
| | | Brightness Temperature (BT), $\nu = 89\,\mathrm{GHz}$, tilted by $25°$, 1-2 s |
| P5 | MiRAC-P | Brightness Temperature (BT), $\nu = 183.31, 243, 340\,\mathrm{GHz}$, nadir view, 1-2 s |
| P5 | Sun Photometer | Spectral Aerosol Optical Depth (AOD) $\lambda = 400 - 2000\,\mathrm{nm}$), 1 s |
| **Aerosol Microphysics** | | |
| P6 | CPC | Number Concentration, $D_\mathrm{p} = 10\,\mathrm{nm} - 3\,\mu\mathrm{m}$, 3 s |
| P6 | PSAP | Absorption Coefficient, $\lambda = 565\,\mathrm{nm}$), 30 s |
| P6 | SP2 | rBC Mass/Number Concentration, PNSD, rBC Mass: $0.26 - 125\,\mathrm{fg}$, $D_\mathrm{p} = 65 - 510\,\mathrm{nm}$, 1 s |
| P6 | UHSAS–1 | Aerosol PNSD, $D_\mathrm{p} = 60\,\mathrm{nm} - 1\,\mu\mathrm{m}$, 3 s |
| P6 | UHSAS–2 | Aerosol PNSD, $D_\mathrm{p} = 80\,\mathrm{nm} - 1\,\mu\mathrm{m}$, 1 s |
| P6 | Grimm Sky-OPC | Aerosol PNSD, $D_\mathrm{p} > 250\,\mathrm{nm}$, 6 s |
| **Cloud Microphysics** | | |
| P6 | PHIPS | Angular Scattering Function, Particle Shape, $D_\mathrm{p} = 20 - 700\,\mu\mathrm{m}$, 20 Hz |
| P6 | SID-3 | Cloud PNSD, Particle Shape, Sub-Micrometer Scale Complexity, $D_\mathrm{p} = 5 - 45\,\mu\mathrm{m}$, 1 Hz |
| P6 | CDP | Cloud PNSD, $D_\mathrm{p} = 2 - 50\,\mu\mathrm{m}$, 1 Hz |
| P6 | CIP | Cloud PNSD, Particle Shape, $D_\mathrm{p} = 75 - 1550\,\mu\mathrm{m}$, 1 Hz |
| P6 | PIP | Precipitation PNSD, $D_\mathrm{p} = 300 - 6200\,\mu\mathrm{m}$, 1 Hz |
| **Aerosol Chemistry** | | |
| P6 | ALABAMA | Single particle composition (Refractory, Non-Refractory),$D_\mathrm{p} = 100 - 1000\,\mathrm{nm}$, 1 Hz |
| **Trace Gas Chemistry** | | |
| P6 | Aerolaser AL5002 | CO-Concentrations, $0 - 100,000\,\mathrm{ppbv}$, 1 Hz |
| P6 | Licor 7200 | $CO_2$ Concentration, $0 - 3000\,\mathrm{ppmv}$, 1 Hz |
| | | $H_2O$ Concentration, $0 - 60\,\mathrm{mmol/mol}$, 1 Hz |
| P6 | 2BTech O3 Monitor | $O_3$-Concentration, $0 - 250\,\mathrm{ppmv}$, 0.5 Hz |

## 2.1 High-frequency wind vector, air temperature, and humidity

On both aircraft, identical sensors were installed in a noseboom for high-frequency measurements of the wind vector and of the air temperature (Hartmann et al., 2018). The basic sensors are an Aventech five-hole-probe placed at the tip of the noseboom and an open-wire Pt100 installed side-wards in a Rosemount housing. All data were recorded and published with a frequency of

5 100 Hz (Hartmann et al., 2019, https://doi.pangaea.de/10.1594/PANGAEA.900880). The response time of the sensors is below 0.01 s, well suited for atmospheric turbulence flux measurements (Lee, 1993). The five-hole-probe is heated during the flight to prevent icing. It is equipped with a purging system to eject water that might have entered the central hole. Thus, measurements within clouds are reliable.

Pressure measurements in the five-hole-probe are recorded by differential pressure transducers of type Setra 239 R for angle

10 of attack, angle of sideslip and the dynamic pressure and by a Setra 278 for the static pressure. To convert the wind vector





measured with respect to the aircraft frame into Earth-fixed coordinates, the position, movement, and attitude of the aircraft is measured by a combination of a high-precision global positioning system (GPS) receiver and an inertial navigation system (INS). The INS, a Honeywell Laseref V, provides longitude, latitude, ground speed, and angular rates and calculates the pitch, roll, and true heading angles with an accuracy of $0.1°$ (roll and pitch) and $0.4°$ (true heading). A Novatel GPS FlexPak6

receiver supports the calculation of the position and the velocity vector. Doppler-derived velocities ("Novatel bestvel") are obtained with a precision of $0.03\,\mathrm{m\,s^{-1}}$. For the final data product, the INS and GPS data were merged by complementary filtering at a frequency of 0.1 Hz.

The wind vector was calculated by applying the procedure described by Hartmann et al. (2018). The method considers a careful calibration of the initial wind measurements, which is based on a combination of the differential measurement capabil-

ities of the GPS and the high-accuracy INS. With the precise aircraft position and attitude, the horizontal wind components are derived with an absolute accuracy of $0.2\,\mathrm{m\,s^{-1}}$ for straight and level flight sections. The vertical wind can only be analyzed as the deviation from the average vertical wind. Therefore, the mean wind vector was averaged for flight sections of at least several kilometers length. For straight and level flight sections, the accuracy of the vertical wind speed relative to the average is about $0.05\,\mathrm{m\,s^{-1}}$.

The temperature measurements were corrected for the adiabatic heating by the dynamic pressure. The absolute accuracy of the temperature measurements is $0.3\,\mathrm{K}$ with a resolution of $0.05\,\mathrm{K}$. The lateral displacement between wind and temperature sensors (radial distance to the center of the five-hole probe of $16\,\mathrm{cm}$ and an axial distance of $35\,\mathrm{cm}$) was found to be not critical. For typical true air speeds of $60\,\mathrm{m\,s^{-1}}$, this axial distance corresponds to a time lag of about $6\cdot10^{-3}\,\mathrm{s}$, which is less than one sample at the recording frequency. Additionally, Polar 5 carried a closed-path LI-7200 gas analyzer for $CO_2$ and

$H_2O$ concentration measurements. The performance of the analyzer with respect to airborne humidity flux measurements has been tested as described in detail by (Lampert et al., 2018). For slow humidity measurements (frequency of 1 Hz), a Vaisala HMT-333, which includes a temperature and HUMICAP humidity sensor, was mounted in a Rosemount housing. Based on the temperature measurements (uncertainty of $0.1\,\mathrm{K}$), the humidity data were corrected for adiabatic heating and reach an accuracy of $0.4\,\%$ (Hartmann et al., 2018).

The achieved accuracy of wind and temperature measurements is sufficient to derive turbulent fluxes of momentum and sensible heat in the atmospheric boundary layer with the eddy-covariance method when straight, horizontal flight sections are analyzed. The majority of measurements during ACLOUD have been obtained over sea ice and in slightly unstable or stable stratification where turbulent heat fluxes are rather small (heat fluxes in the order of a few $\mathrm{W\,m^{-2}}$ for flight legs of about $10\,\mathrm{km}$ length). Such low flux conditions represent a challenge for the instrumentation because deviations of both wind

and temperature from the average values are small. This results in larger relative uncertainties of the derived turbulent fluxes compared to a more convective environment in cold air outbreaks over open water. Therefore, the vertical profiles of turbulent fluxes were calculated from staircase flight patterns of at least $10\,\mathrm{km}$ averaging length, which reduces the uncertainties. As shown in Wendisch et al. (2019) (Fig. 18), the derived profiles are in agreement with theory showing downward heat fluxes in stable environment and upward fluxes in a well-mixed surface forced convective layer and are capable to reveal the impact



of clouds and of the surface on the turbulent fluxes. Nevertheless, the interpretation of the turbulent fluxes in future studies requires a careful consideration of the entire meteorological situation.

## 2.2 Spectral solar radiation

Spectral solar radiation was measured by three different instruments on board of Polar 5. The Spectral Modular Airborne
Radiation measurement sysTem (SMART-Albedometer) primarily measures upward and downward spectral solar irradiances in the wavelength range between 350 nm and 2200 nm (Wendisch et al., 2001; Ehrlich et al., 2008; Bierwirth et al., 2013). Additionally, upward radiances are obtained for wavelengths below 1000 nm with optical inlets covering a 2.1° field of view (FOV). All optical inlets are actively horizontally stabilized to correct for changes of the aircraft attitude of up to 6° with an accuracy of 0.2° (Wendisch et al., 2001). Two types of grating spectrometers are applied by the SMART-Albedometer.
At wavelengths below 1000 nm, the spectrometers provide a spectral resolution of 2–3 nm. The near-infrared spectrometers (950–2200 nm) have a coarser resolution of 12–15 nm. For these near-infrared spectrometers, the raw data were corrected for the dark signal using regular dark measurements with opto-mechanical shutters. The spectrometers measuring below 1000 nm wavelength register the dark signal by integrated dark reference pixels. All quantities measured by the SMART-Albedometer were merged and published in a combined data set (Jäkel et al., 2019, https://doi.org/10.1594/PANGAEA.899177).

The Airborne Imaging Spectrometer for Applications (AISA) Eagle/Hawk (two pushbroom hyperspectral imaging spectrometers operated in tandem) observes two-dimensional (2D) fields of upward spectral solar radiance (Schäfer et al., 2013, 2015). Each of the two components consists of a single-line sensor with 1024 (AISA Eagle) and 384 (AISA Hawk) spatial pixels, respectively. The spatial resolution (cross-track pixel sizes) of the AISA Eagle/Hawk measurements is in the order of 4 m for a cloud situated 2 km below the aircraft. For each spatial pixel, the wavelength range of 400–2500 nm is spectrally
resolved. The dark signal correction is obtained automatically by an integrated shutter. The measurements of AISA Eagle and AISA Hawk were filtered for straight flight legs and published separately to remain the full spatial resolution of both sensors (Ruiz-Donoso et al., 2019, https://doi.pangaea.de/10.1594/PANGAEA.902150).

A digital CANON camera equipped with a downward-looking 180° fish-eye lens measured the directional distribution of upward radiance of the entire lower hemisphere every six seconds (Ehrlich et al., 2012). A Complementary Metal Oxide
Semiconductor (CMOS) image sensor covers the three spectral channels (RGB) centered at wavelength of 591 nm (red), 530 nm (green), and 446 nm (blue) with about 80 nm full-width of half-maximum (FWHM) spectral resolution. The 3908 × 2600 pixel sensor provides an angular resolution of less than about 0.1°. Images were recorded in raw data format to gain the full dynamic range (14 bit) of the camera sensor chip. The processing of the raw data was applied without white balance by setting the multipliers of all channels to 1 (Ehrlich et al., 2012). The dark signal of the images was quantified in the laboratory
for different camera settings and does not exceed one digital unit of the 15 bit dynamic range. An identical digital camera system was installed on Polar 6. So far, only the measurements on Polar 5 have been processed and published (Jäkel and Ehrlich, 2019, https://doi.org/10.1594/PANGAEA.901024).

All three systems were radiometrically, spectrally, and geometrically calibrated in the laboratory. A 1000-Watt standard calibration lamp (traceable to the standards of the National Institute of Standards and Technology, NIST) was applied for

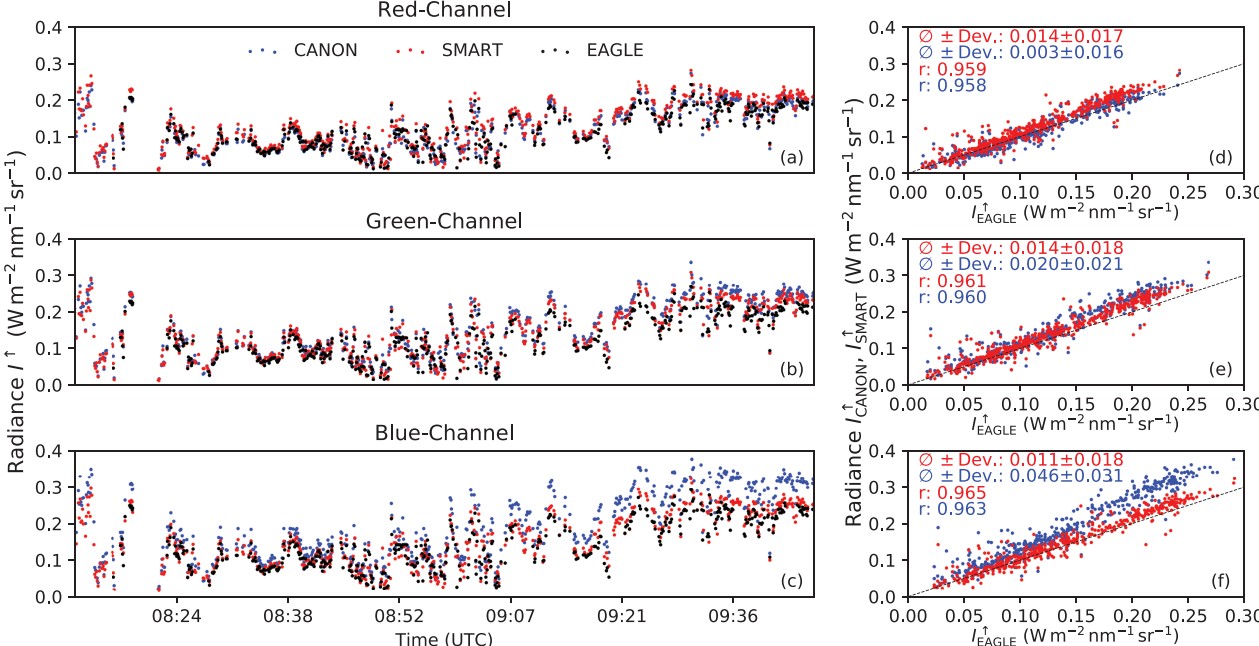

**Figure 1.** Comparison of spectral radiance in nadir direction $I^{\uparrow}$ measured by SMART, AISA Eagle/Hawk, and the CANON fish-eye camera on 27 May 2017 (Flight #6). All data are convoluted to the three spectral bands of the fish-eye camera. Time series for all bands (Panel a) and scatter plots using the radiance of AISA Eagle/Hawk as reference (Panel b) are shown. $\varnothing$ gives the mean and "Dev" the standard deviation of the difference between $I^{\uparrow}$ measured on Polar 5 and 6. $r$ denotes the Pearson's correlation coefficient.

the irradiance measurements of the SMART-Albedometer. All radiance measurements were calibrated with the same NIST traceable radiance source (integrating sphere). In-field calibrations with a secondary-calibrated integrating sphere were used to track and correct systematic changes of the calibrations which may appear during the integration on the aircraft.

The total uncertainties of the radiance measurements mostly originate from the radiometric calibration given by the uncertainty of the applied radiation source and the signal to noise ratio that differs with wavelength due to the sensitivity of the sensors. Assuming typical measurements above clouds or snow, the uncertainties of upward radiance measured by the SMART-Albedometer range between 6 % at wavelengths below 1000 nm and 10 % for longer wavelengths. For the irradiance measurements of the SMART-Albedometer similar uncertainties are given by Bierwirth et al. (2009).

The calibration of all three systems was verified by comparing the upward radiances measured in nadir direction. The spectrally higher resolved measurements by the SMART-Albedometer and the AISA Eagle/Hawk were convoluted to the three spectral bands of the fish-eye camera (Ehrlich et al., 2012). Figure 1 shows a time series of the three spectral bands for a two hour flight section of 27 May 2017 (Flight #6) and the corresponding scatter plots using AISA Eagle/Hawk as reference. To match the same 2.1° nadir spot of the SMART-Albedometer, measurements of AISA Eagle/Hawk and the 180° fish-eye camera have been corrected for the aircraft attitude. For AISA Eagle/Hawk the 57 center pixel were averaged over



ten time steps. For the 180° fish-eye camera the 2.1° nadir spot is covered by 1177 spatial pixel. The comparison is limited to measurements where the aircraft did not exceed a horizontal misalignment of roll or pitch angle more than 2°. The time series covers clouds of different reflectivity and shows agreement between all three sensors in the observed dynamic range. The time series (Figure 1a,b,c) show that all instruments captured the general cloud structure. Differences occur only on small temporal

scales, likely due to the slightly different field of view and the different integration times which range between 500 ms for the SMART-Albedometer, 30 ms for AISA Eagle/Hawk, and 0.6 ms for the fish-eye camera. The regression of the radiances of SMART-Albedometer and AISA Eagle/Hawk and (red dots in Fig.1 d,e,f) shows a small offset of less than 1%, which is similar to previous measurement campaigns (Bierwirth et al., 2013; Ehrlich et al., 2012). While the red and green channel of the fish-eye camera (blue dots in Fig.1 d,e,f) are comparable to the AISA Eagle/Hawk, a difference is observed for the blue channel.

This comparison of the three instruments was used to inter-calibrate the fish-eye camera in order to provide a consistent data set.

### 2.3  Broadband solar and terrestrial radiation

Upward and downward broadband irradiances were measured by pairs of CMP 22 pyranometers and CGR4 pyrgeometers, covering the solar (0.2-3.6 μm) and thermal-infrared (4.5-42 μm) wavelength range, respectively. Both aircraft, Polar 5 and

6, were configured with an identical set of instruments and sampled with a frequency of 20 Hz. In stationary operation, the uncertainty of the sensors is less than 3 % as characterized by the calibration of the manufacturer and evaluated by, e.g., Gröbner et al. (2014). For the airborne operation of the fixed-mounted sensors, the misalignment of the aircraft was corrected by applying the approach by Bannehr and Schwiesow (1993), and Boers et al. (1998), which was applied for the downward direct solar irradiance. Therefore, the fraction of direct solar radiation was estimated using radiative transfer simulations (cloud

free and cloud covered). The simulations were based on available in-flight observations and consider the temperature and humidity profiles and cloud cover. In case of clouds, the cloud optical thickness was fixed to a representative value of 5. The upward solar radiation as well as the upward and downward terrestrial radiation were assumed to be isotropic and were not corrected for the aircraft attitude. To limit the remaining uncertainties due to the aircraft movement, measurements with roll and pitch angles exceeding ±4° were removed from the data set.

To account for the slow response of the pyranometer and pyrgeometer, a correction of the instrument inertia time following the approach by Ehrlich and Wendisch (2015) was applied. Response times of 2 s and 6 s (e-folding time), characterized in laboratory measurements, were applied for the pyranometer and pyrgeometer measurements. Assuming a typical ground speed of 60 m s$^{-1}$ and a flight altitude of 100 m, the correction enables to reconstruct horizontal fluctuations up to scales of 3 m.

During flights inside clouds, icing by super-cooled liquid water droplets might have effected the radiation measurements

after ascents and descents through the clouds. Using on-board video camera observations, the data were screened for icing events when the solar downward irradiance appeared artificially reduced. As this detection of icing was not always reliable, uncertainties remain.

Surface brightness temperature was measured by a nadir looking Kelvin infrared radiation Thermometer (KT–19). These measurements were converted into surface temperature values assuming an emissivity of 1. This is justified due to the small

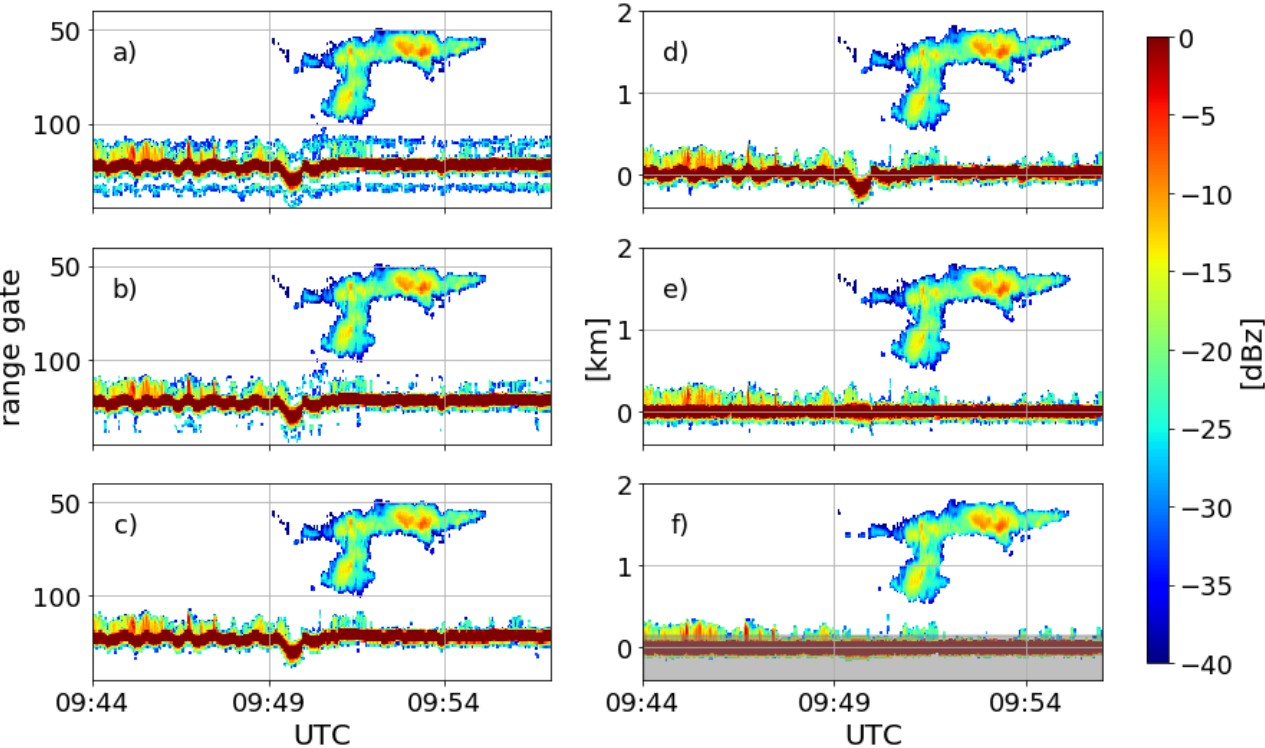

**Figure 2.** Time series of radar reflectivity profiles measured on 25 May 2017 (flight #23) for different processing steps: a) raw data, b) after subtraction of mirror signal, c) after speckle filter, d) filtered data on a time-height grid, e) corrected for sensor altitude, mounting position, pitch and roll angle, f) remapping onto a constant vertical grid. The grey shading indicates the range of surface contamination (≤150 m

impact of atmospheric absorption in the wavelength range of 9.6 μm to 11.5 μm for which the KT-19 is sensitive (Hori et al., 2006). With a sampling frequency of 20 Hz, the KT-19 resolves small scales of the surface temperature heterogeneities such as observed in case of leads in sea ice (Haggerty et al., 2003). The processed data of the KT-19, pyranometer and pyrgeometer were merged and published in a combined data set (Stapf et al., 2019, https://doi.org/10.1594/PANGAEA.900442).

## 2.4 Active and passive microwave remote sensing

The Microwave Radar/Radiometer for Arctic Clouds (MiRAC;  Mech et al., 2019) has been designed for operation on board of Polar 5. It consists of a single vertically polarized frequency modulated continuous wave (FMCWF) cloud radar RPG-FMCW-94-SP including a passive channel at 89 GHz (MiRAC-A) and a microwave radiometer (MiRAC-P) with six channels along the strong water vapor absorption line at 183.31 GHz and two window channels at 243 GHz and 340 GHz. MiRAC-A is operated in a bellypod fixed below the aircraft fuselage pointing about 25° off nadir (along track backwards), while MiRAC-P is integrated in the cabin pointing nadir. The cloud radar of MiRAC-A provides vertically resolved profiles of the equivalent radar reflectivity. The vertical resolution depends on the chirp sequences and the temporal resolution which varied





between 1-2 s. During ACLOUD three different settings with resolutions between 4 m and 30 m were used. A multi-step processing of the radar data was performed to correct disturbances in radar signal due to the strong surface return and to convert them into geo-referenced data taking the sensor mounting and the aircraft position into account (Mech et al., 2019). Fig. 2 illustrates the effect of the processing steps, which finally lead to regularly gridded data which become reliable 150 m above

ground level. The passive channels receive microwave emission from the surface and the atmosphere. The 89 GHz channel is especially sensitive to the surface emission with high brightness temperatures measured above sea ice. The lower emission over sea ice allows for retrieving information on the liquid water path. The channels around the 183.31 GHz sense atmospheric moisture, which successively stems from lower layers when the distance to the center of the emission line increases. With increasing frequency, larger snow particles can lead to a brightness temperature depression due to scattering effects. The

processed data of MiRAC-A and Mirac-P were merged and published in a combined data set (Kliesch and Mech, 2019, https://doi.org/10.1594/PANGAEA.899565).

## 2.5 Remote sensing by lidar

The active microwave profiling by MiRAC was complemented by the Airborne Mobile Aerosol Lidar (AMALi) system (Stachlewska et al., 2010). This backscatter lidar has three channels: one unpolarised channel in the ultraviolet (UV) at 355 nm and

two channels in the visible spectral range at 532 nm (perpendicular and a parallel polarised). AMALi consists of a Nd:YAG laser, a receiving optical system including the telescope, and opto-electrical components converting the backscattered light intensities into digital data. Overlap between the transmitted laser beam and the receiving telescope is achieved for ranges larger than 235 nm (Stachlewska et al., 2010). Data are recorded with 7.5 m range and 1 s temporal resolution, which yields a horizontal resolution of 75 m for typical aircraft speeds over ground of $270 \, \mathrm{km \, h^{-1}}$. To improve the signal-to-noise ratio,

temporal averaging was performed.

During ACLOUD, AMALi was installed pointing downwards (except on Flight #10 where it pointed into zenith direction) through a floor opening of Polar 5, thus, probing the atmosphere between the flight level and the surface. For eye safety reasons, AMALi was operated at flight levels above 2700 m only. The data processing eliminated the background signal which mainly results from scattered sun light and electronic noise. Additionally, a drift of the so-called base line of each channel was corrected

for. The backscattered intensities depend on the amount and electromagnetic properties of the scatterers. Neglecting aerosol extinction, the attenuated backscatter coefficients for each channel were calculated from the background-corrected signals by normalizing the measurements to a typical air density profile (Stachlewska et al., 2005). For the ACLOUD campaign data from the AWIPEV station in Ny-Ålesund have been used (Maturilli, 2017a, b). From the attenuated backscatter coefficients at the 532 nm the depolarisation ratio was calculated by dividing the values from the perpendicular and parallel channels. This ratio

provides qualitative information on the shape of the scatterers. High values indicate non-spherical scatterers like ice crystals. Making use of the two wavelength channels, the colour ratio was calculated as the ratio of the attenuated backscatter coefficients of the 355 nm and the 532 nm parallel channel indicating the size distribution of the aerosol load. Clouds below the aircraft were identified in the 532 nm parallel channel by the strength of their attenuated backscatter signal. They were discriminated from aerosols by using a threshold value of five-times the reference value of a cloud free section of the backscatter profiles.



Cloud top height was determined as the highest altitude, which meets the above criterion for consecutive altitude bins. In the published data set (Neuber et al., 2019, https://doi.org/10.1594/PANGAEA.899962), cloud tops in close distance to the aircraft (less then 100 m below the flight level) and low clouds (below 30 m above the ground) are excluded. For consistency to the radar profiles, the AMALi data were converted into "altitude above sea level" by using the GPS altitude.

## 2.6  Sun-photometer

The airborne Sun photometer with an active tracking system (SPTA) was installed under a quartz dome of Polar 5 to derive the spectral aerosol optical depth (AOD). To measure the direct solar radiation as well as solar radiance, the optics of the SPTA use a sunspot which is focused via a diaphragm with a field of view of $1°$. It operates a filter wheel with 10 selected wavelengths in the spectral range from 367 nm to 1024 nm. With knowledge of the extraterrestrial signal the spectral optical depth of the atmosphere as well as spectral optical depth of aerosol was derived. The algorithm applied for the SPTA is based on Herber et al. (2002). The extraterrestrial signal was calculated based on a Langley Calibration, which are performed regularly in the high mountain area (Izana, Tenerife). The data was screened for contamination by clouds to minimize an artificial enhancement of the AOD by thin clouds. The cloud screening algorithm applied a threshold of measured irradiance and made use of the higher temporal/spatial variability of clouds compared to the rather smooth changes of aerosols properties (Stone et al., 2010).

## 2.7  Thermodynamic sounding

The Advanced Vertical Atmospheric Profiling System (AVAPS) was operated on Polar 5 to release dropsondes of type RS904 (Ikonen et al., 2010). The sondes measure vertical profiles of air temperature, humidity, pressure, and the horizontal wind vector between the flight altitude of typically 3-4 km and the surface. The vertical resolution of the profiles is about 5 m determined by the fall velocity of about $10\,\mathrm{m\,s^{-1}}$ and the sampling frequency of 2 Hz. The Atmospheric Sounding Processing Environment (ASPEN) software package was used to correct the raw data for the slow time response of the temperature sensor and to remove the known humidity bias (Voemel et al., 2016). Data close to the aircraft, where the sensors did not jet adjust to the outside temperature, and invalid measurements were removed by the quality check of ASPEN. To resolve fast temperature and humidity changes at cloud top, the time response of the sensors has been corrected by an alternative method following Miloshevich et al. (2004). A time response (e-folding) of 4 s was applied to the temperature sensor and 5 s to the humidity sensor. Both data, processed by ASPEN and additionally corrected for the time response using the approach by Miloshevich et al. (2004), are included in the published data set (Ehrlich et al., 2019a, https://doi.org/10.1594/PANGAEA.900204).

## 3  Instrumentation of Polar 6

Polar 6 was primarily equipped with in situ instruments characterizing aerosol particles, cloud droplets, ice crystals, and trace gases (Table 2). Cloud particles were sampled with five different optical array and scattering probes. Using a counterflow virtual impactor (CVI), the aerosol particles and cloud particle residuals were collected and characterized by the in-situ aerosol





instrumentation. The trace gas instrumentation measured concentrations of CO, $CO_2$, $O_3$ and water vapor. Meteorological properties including turbulent and radiative fluxes were measured with an instrumentation identical to that operated on Polar 5.

## 3.1 Cloud particle in situ measurements

Four wing pylons are available on Polar 6, two on each wing. For ACLOUD five different probes have been installed to sample
cloud particle microphysical and optical properties: the Cloud Droplet Probe (CDP), the Cloud Imaging Probe (CIP), the Precipitation Imaging Probe (PIP), the Small Ice Detector Mark 3 (SID-3), and the Particle Habit Imaging and Polar Scattering (PHIPS). Two configurations were applied. The combination of PIP, CIP, SID-3, and PHIPS was operated during the first half of ALCOUD (Flights #8-15). In the second half (Flights #16-24), the PIP was replaced by the CDP to improve the sampling of small cloud droplets, which dominated the rather warm clouds observed during ACLOUD.

### 3.1.1 Cloud Droplet Probe

The Cloud Droplet Probe (CDP) is a forward scattering optical spectrometer (size range 2-50 μm) using a single mode diode laser at a wavelength of 0.658 μm (Lance et al., 2010; Wendisch et al., 1996). The instrument counts and sizes individual droplets by detecting pulses of light scattered from a laser beam in the near-forward direction (4-12°). Sizes are accumulated in 30 bins with variable widths. For ACLOUD, a 1 μm bin-width was chosen for small droplet sizes (2-14 μm), while larger cloud
droplets (16-50 μm) were collected in 2 μm bins. The particle diameter was deduced from the measurement using a scattering cross section to diameter relationship based on the Mie Theory. This relationship is a non monotonic function, which can give multiple solutions for one scattering cross section measurement. Therefore, the particle number size distribution (PNSD) was obtained in two steps. First, the CDP raw PNSD was computed by the probe manufacturer software, which uses the first solution of the Mie theory particle size determination. This PNSD has then been corrected using a Monte Carlo inversion method to
ensure equiprobable values to all possible solutions of the Mie theory particle size determination .In order to do so, the particle counts ($N_{\mathrm{raw}}$) from one raw size bin were uniformly distributed into a finer binning ($N_{\mathrm{fine}}$) for a more precise particle size determination and a scattering cross section was computed for each $N_{\mathrm{fine}}$. A diameter was then randomly attributed to each counts of $N_{\mathrm{fine}}$ using the different solution given by the Mie theory with equiprobability and these diameters were distributed into the same original size bins ($N_{\mathrm{cor}}$). Prior to its use, the probe has been calibrated using glass beads for sizing and a single
droplet generator (Lance et al., 2010; Wendisch et al., 1996) for the sample area (0.32 mm$^2$). The CDP is equipped with anti-shatter tips to reduce possible shattering artifacts (Korolev et al., 2011) and allows the retrieval of particle by particle information. Microphysical quantities such as liquid water content ($LWC$) and effective droplet diameter $D_{\mathrm{eff}}$ were derived from the PNSD.

### 3.1.2 Cloud Imaging Probe and Precipitation Imaging Probe

The Cloud Imaging Probe (CIP) and Precipitation Imaging Probe (PIP) measure the size and the shape of cloud particles (Baumgardner et al., 2011). Their measurement principle is based on that of Optical Array Probes (OAPs, Knollenberg,





1976), which use the linear array technique to acquire two-dimensional black and white images of particles. As the particles pass through the laser they cast a shadow, which is recorded on a photodiode array and analyzed for particle dimension and shape. According to the resolution of the photodiode and their quantity, the CIP and PIP have nominal size ranges of 25-1550 µm (25 µm resolution and 64 diodes) and 100-6200 µm (100 µm resolution and 64 diodes), respectively. The particle size

distribution of hydrometeors are computed from the OAP images. The assessment of the Median Mass Diameter ($MMD$) and the Ice Water Content ($IWC$) relies on the definition of the crystal diameter and its mass-diameter relationship. Two mass-diameter relationships were considered in the data set: Baker and Lawson (2006) denoted with BL06, and Brown and Francis (1995) labelled with BF95. Following the approach by Crosier et al. (2011), non-spherical ice crystals were separated from liquid droplets based on their circularity parameter (circularity larger than 1.25 and image area larger than 16 pixels). Only these

non-spherical particles images were used for the computation of the "ice" phase. Possible contamination of shattering/splashing of ice/liquid particles on the instruments tips have been identified and removed using inter arrival time statistics and image processing (Field et al. 2006). Due to the large OAP measurement uncertainties for the smallest sizes, the first two PNSD size bins were removed. A complete description of the data processing including a discussion of the applied mass-diameter relationships can be found in Leroy et al. (2016) and Mioche et al. (2017).

In the CDP, CIP, and PIP data set published in the PANGAEA database (Dupuy et al., 2019, https://doi.pangaea.de/10.1594/PANGAEA.899074), the PSDNs of all instruments are stored separately. In order to retrieve the most statistically reliable PSDN, every particle images were used (suffix "ALL"). Truncated images were extrapolated in order to estimate the particle diameter following Korolev et al. (2000). However, the classification of non-spherical particles was based on complete images only (suffix "ALL-IN"). Depending on the application, different definitions of the particle diameters can be applied when

calculating the PSDN. This is why three PSDNs are provided, each based on one of three different diameters ($D_{\mathrm{max}}$, $D_{\mathrm{eq}}$ and $D_{\mathrm{cc}}$) which are defined as:

– $D_{\mathrm{max}}$ or length is the maximum dimension originating from the image center of gravity (see Leroy et al., 2016). It was used in previous studies in the region (Jourdan et al., 2010).

– $D_{\mathrm{eq}}$ or equivalent diameter is the diameter of the circle which has the same surface as the particle image. Vaillant de

Guélis et al. (2018) show that it is the least subjected to error in sizing due to out of focus deformation of the image. Also, as it represents a surface, its property is closer to the scattering cross section and thus more comparable to the CDP measurements.

– $D_{\mathrm{cc}}$ or circumpolar diameter is the diameter of the circle encompassing the particle image. This is the diameter used in the BF95 mass diameter relationship.

### 3.1.3  Small Ice Detector

The Small Ice Detector Mark 3 (SID-3) records the spatial distribution of the forward-scattered light from single cloud particles in the angular region of 5° to 26° as 2D scattering patterns (Hirst et al., 2001). Cloud particles passing a laser beam (wavelength 532 nm) are detected using two nested trigger optics that have circular apertures with a half angle of 9.25° located at ±50°





relative to the forward direction. The maximum camera acquisition rate is 30 Hz, whereas the trigger detector has a maximum acquisition rate of 11 kHz. The trigger signal is recorded as a histogram that can be used to retrieve the cloud particle size distribution using size calibration procedures described in Vochezer et al. (2016). The PNSD covers a size range of 5-45 μm divided into 16 size bins (2-5 μm resolution). From a sub-sample of the detected particles, a high resolution 2D scattering
pattern is acquired. These scattering patterns were analyzed for the particle shape and sphericity using methods described in Vochezer et al. (2016) or for the particle mesoscopic complexity using the methods described in Schnaiter et al. (2016). The SID-3 data sets available in PANGAEA contain 1 Hz particle PNSD (Schnaiter and Järvinen, 2019a, https://doi.org/10.1594/PANGAEA.900261) and the analysis results of the individual 2D scattering patterns (Schnaiter and Järvinen, 2019b, https://doi.org/10.1594/PANGAEA.900380). For each detected particle, information of the particle sphericity, shape and mesoscopic
crystal complexity are given.

### 3.1.4   Particle Habit Imaging and Polar Scattering

The Particle Habit Imaging and Polar Scattering (PHIPS) probe is a combination of a polar nephelometer and a stereoscopic imager (Abdelmonem et al., 2016; Schnaiter et al., 2018). The two parts of the instrument are combined by a trigger detector so that both imaging and scattering measurements are performed on the same single particle. The polar nephelometer has 20
channels from 18° to 170° with an angular resolution of 8° recording single particle angular scattering functions. The stereo-microscopic imager consists of two camera and microscope assemblies with an angular viewing distance of 120° acquiring a bright field stereo-microscopic image. The magnification of the microscopes can be varied in the range from 1.4x to 9x, which corresponds to field of view dimensions ranging from $6.27 \times 4.72\,\mathrm{mm}^2$ to $0.98 \times 0.73\,\mathrm{mm}^2$, respectively. The optical resolution at the highest magnification setting is about 2.3 μm. During ACLOUD two different magnifications of 6x and 8x were set for
the two PHIPS microscopes of camera 1 and 2, respectively. The purpose of this setting is to capture a detailed view of the particle in camera 2 while ensuring that the same particle was completely captured by camera 1. Particles that were completely captured within the field of view of either camera were analyzed for their size, sphericity and position within the image as explained in Schön et al. (2011). Furthermore, the images were manually assigned to different shape classes. The PHIPS data set available in PANGAEA contains separate image overviews for both cameras per flight. Further, it contains single-particle
angular light scattering data for each recorded particle. For a sub-sample of particles, the microphysical information derived from the image analysis were combined in a single ASCII file per flight.

### 3.1.5   Combined cloud particle number size distributions

When flown together (Flight #16-26), CDP, SID-3, and CIP data can be combined for merged PNSDs that cover a size range between 2 μm and 1550 μm. Figure 3 shows PNSD of all instruments averaged over the entire flight of 18 June 2017 (Flight #20).
Only data with liquid water content above $1\,\mathrm{mg\,m^{-3}}$ were included. For the CDP, also the uncorrected PNSD produced by the manufacturer software was included, which shows a significant overestimation of small droplet below 8 μm compared to the corrected version. The PNSD derived from SID-3 measurements agrees well with the CDP and both match the smallest bins of the CIP. For CIP, all three options to calculate the particle diameter are presented. The choice of diameters definition mostly

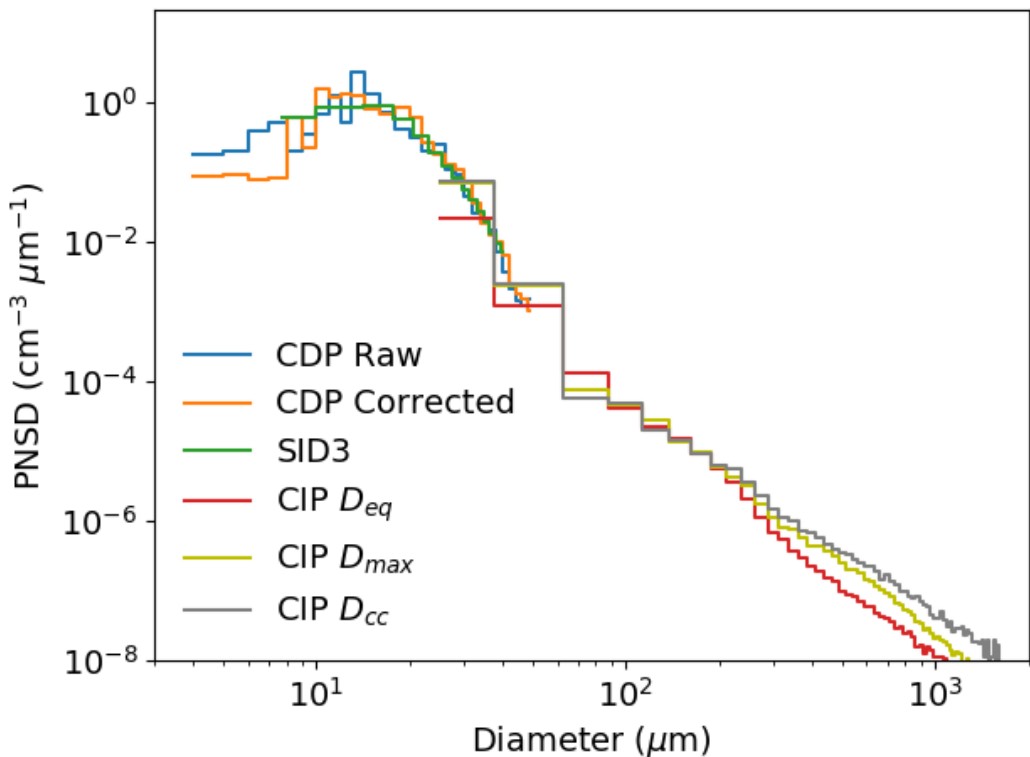

**Figure 3.** Comparison of averaged PNSD derived from CPD, SID-3 and CIP during Flight #20 on 18 June 2017. For CDP the corrected and uncorrected PNSD produced by the manufacturer software are shown. For CIP, all three options to calculate the particle diameter are presented.

affects ice crystals larger than 200 μm where the equivalent diameter $D_{eq}$ gives the lowest (assuming smaller crystals) and the circumpolar diameter $D_{cc}$ the highest ice crystal concentrations (assuming larger crystals).

### 3.2 Aerosol particle measurements

Ambient aerosol particles and cloud particle residuals were collected by two inlets on board Polar 6. Their microphysical and
5  chemical properties were measured inside the cabin by a suite of aerosol sensors (Table 2). A third and fourth inlet provided ambient air for the in-cabin instrumentation of trace gas analysis. The characteristics and the handling of the different inlets is discussed below in Section 3.4.

#### 3.2.1 Aerosol particle number concentration and number size distribution

Two ultra-high sensitivity aerosol spectrometer (UHSAS CPC, Cai et al., 2008) were operated either at different inlets (for
10  simultaneous measurements) or at the same inlet (for inter-comparison). The flow rate was set to $50\,\mathrm{ml\,min^{-1}}$. The UHSAS



measures the number size distribution of particles with diameters between 60 nm and 1000 nm by detecting scattered laser light divided in 100 user-specified size bins of variable size (2-30 nm resolution). From these measurements, the mean particle diameter and the particle number concentration of a defined size range were derived. From the data evaluation it was inferred that the UHSAS-1 and the UHSAS-2 could reliably detect particles larger than 60 nm and 80 nm, respectively. During ACLOUD,

the UHSAS-1 broke during Flight #19 (17 June 2017 around 12:00 UTC), i.e. from this moment only the UHSAS-2 could be used for the scientific analysis.

The calibration of both UHSAS were compared during flights when both instruments were connected to the same particle inlet. Figure 4 shows the total particle concentration (80-1000 nm) and averaged particle number size distributions from Flights #7, #14, and #18. The PNSDs of both instruments (Figure 4b) match in the entire size range for all three flights. For

Flights #14 and #18, the total particle concentration (Figure 4a) of UHSAS-1 was found to be about 8 % higher than measured by UHSAS-2 while on Flight #7 no significant difference could be observed.

Additionally to the UHSAS, an optical particle counter (OPC Grimm 1.129) was operated to measure size distribution and number concentration of particles larger than 250 nm in diameter. Due to sampling line losses in the aerosol inlet and in the CVI the upper size limit of the OPC was estimated to about 5 μm. A condensation particle counter (TSI-3010 CPC Mertes et al.,

1995) measured the total particle number concentration by a light scattering technique after creating aerosol droplets inside the instrument large enough for detection. This way, particles down to diameters of 10 nm and up to 3 μm were measured at a sample flow of $1\,l\,min^{-1}$. The measurements of the UHSAS-1, the CPC, and parameters of the CVI operation were merged and published in a combined data set (Mertes et al., 2019, https://doi.pangaea.de/10.1594/PANGAEA.900403). To provide the full sampling frequencies, the UHSAS-2 data (Zanatta and Herber, 2019a, https://doi.org/10.1594/PANGAEA.900341)

and the OPC measurements (Eppers and Schneider, 2019a, https://doi.pangaea.de/10.1594/PANGAEA.901149) are published separately.

### 3.2.2 Light absorbing particles

The absorption coefficient of the sampled particles was measured by a single-wavelength particle soot absorption photometer (TSI-3010 PSAP, Bond et al., 1999) with a time resolution of 30 s. The PSAP uses the filter based integrated plate technique

in which the change in optical transmission caused by particle deposition is related to the optical absorption coefficient. To calculate the absorption coefficient the correction given in Bond et al. (1999) Eq. 12 was applied. Only the correction term including the scattering coefficient was neglected, because particle scattering was not measured. However, since the filters were changed when the transmittance was still high, the scattering correction is of minor importance. In order to calculate BC mass concentrations a mass absorption cross section of $10\,m^2\,g^{-1}$ was used. Assuming a mass absorption cross-section the

absorption coefficient can be transferred into a mass concentration of equivalent black carbon (e.g,. Mertes et al., 2004). All measurements of the PSAP are included in the CVI data set (Mertes et al., 2019, https://doi.pangaea.de/10.1594/PANGAEA. 900403).

The single particle soot photometer (SP2, Stephens et al., 2003) was used to quantify the concentration and size distribution of refractory black carbon (rBC). Briefly, the SP2 is based on the laser-induced incandescence technique that allows quantifying



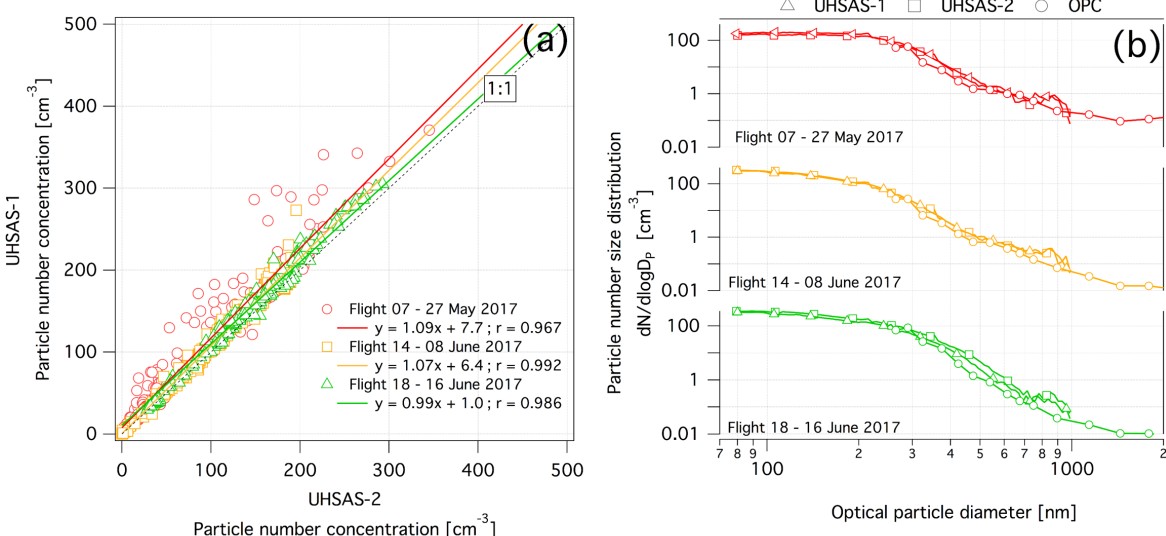

**Figure 4.** Comparison of UHSAS-1 and UHSAS-2 measuring at the same inlet for Flights #7, #14, and #18. Panel a shows integrated PNSD for a size range 80-1000 nm. In panel b, PNSD averaged over the entire flights are compared.

the mass of refractory BC particles despite the presence of other non-absorbing and non-refractory components. The calibration of the incandescence and scattering signal was performed using size selected fullerene soot particulate (Alfa Aesar, stock #40971, lot #FS12S011) and polystyrene latex (Thermo Scientific). A complete description of the calibration set-up, standard materials and operative principles is given by Moteki and Kondo (2010), Gysel et al. (2011), Baumgardner et al. (2012), and Laborde et al. (2012b, a). The number and mass size distribution and the number concentration and mass concentration of rBC particles were obtained for the rBC cores having a mass between 0.40 fg and 187 fg. The rBC core size is commonly expressed as rBC mass equivalent diameter (DrBC), calculated using a void-free material density of 1800 kg m$^{-3}$ (Moteki et al., 2010), the resulting diameter detection range was 70-584 nm. Due to a failure of the scattering detector, the quantification of coating thickness was not possible during ACLOUD. The data set published in PANGAEA (Zanatta and Herber, 2019b, https://doi.org/10.1594/PANGAEA.899937) includes only number and mass concentration of rBC. The concentrations were to low to provide meaningful time series of the size distributions. Averaging of at least 3-4 min outside clouds was required but is still not sufficient in cloud measurements. Data can be requested by contacting the corresponding author of the data set.

### 3.2.3 Chemical particle composition

Aerosol particle and cloud particle residual composition was measured by the Aircraft-based Laser ABlation Aerosol MAss spectrometer (ALABAMA, Brands et al., 2011; Köllner et al., 2017). In the depressurized part of the instrument, particles are detected by two detection lasers. The time of flight between the two laser beems is used to measure the velocity of the particles and to calculate their vacuum aerodynamic diameter. The detected particles are ablated and ionized by a single laser



pulse and the formed ions are analyzed by a bipolar time-of-flight mass spectrometer providing information on single particle chemical composition. The analyzed particle mass spectra (mass-to-charge ratio $m/z$) were assigned to specific particle types by grouping similar mass spectra to clusters, using known marker ions, and comparing to reference mass spectra. Compared to previous missions, the inlet system of the ALABAMA was improved for ACLOUD, such that aerosol particles and cloud

particle residuals in a size range between 200 nm and 1900 nm were sampled. The data showed, that 99 % of the chemically analyzed particles ranged between 250 nm and 1600 nm diameter. To provide the full ion information for each particle, only spectra with significant positive and negative ion signals were considered. During ACLOUD, 245,427 particles in total were chemically analyzed by the ALABAMA (198,256 ambient aerosol particles and 47,171 cloud particle residuals). In a first step, the measured spectra were checked for frequent ion signal peaks and peak combinations. By comparison with known

ion marker peaks from the literature (e.g., Köllner et al., 2017; Pratt and Prather, 2010), eleven different species were defined for the database in this study which are listed in Table 3. Based on these marker peaks external and internal mixtures of the different species were analyzed and grouped into different particle types. The data set published in PANGAEA (Eppers and Schneider, 2019b, https://doi.pangaea.de/10.1594/PANGAEA.901047) provides the chemical composition (particle species) of each individual particle. If available, the particle size defined by the vacuum aerodynamic diameter was added.

**Table 3.** Particle species classification defined by the ion marker peaks observed in the mass spectra (mass-to-charge ratio $m/z$) measured by ALABAMA.

| Particle species containing | Ion marker peaks of mass-to-charge ratio $m/z$ |
| --- | --- |
| Ammonium | $+18$ for $NH_4^+$ |
| Dust | $+40$ for $Ca^+$ or $MgO^+$; $+56$ for $Fe^+$, $CaO^+$, $Si_2^+$ or $MgO_2^+$; $+57$ for $CaOH^+$; $-44$ for $SiO^-$; $-60$ for $SiO_2^-$; $-76$ for $SiO_3^-$ |
| Elemental carbon | $+12 \cdot n$ for $C_n^+$; $-12 \cdot n$ for $C_n^-$; $(n = 1, 2, ...8)$ |
| Levoglucosan | $-45$ for $CHO_2^-$; $-59$ for $C_2H_3O_2^-$; $-71$ for $C_3H_3O_2^-$ |
| Nitrate | $-46$ for $NO_2^-$; $-62$ for $NO_3^-$ |
| Nitrogen containing organics | $-26$ for $CN^-$; $-42$ for $CNO^-$ |
| Potassium | $+39$ and $+41$ for $K^+$ |
| Sodium chloride | $+23$ for $Na^+$; $+81$ and $+83$ for $Na_2Cl^+$ $-35$ and $-37$ for $Cl^-$; $-93$ and $-95$ for $NaCl_2^-$ |
| Sulfate | $-96$ for $SO_4^-$; $-97$ for $HSO_4^-$ |
| Triethylamine (tentatively) | $+86$ for $C_5H_{12}N^+$ |
| Trimethylamine | $+58$ for $C_3H_8N^+$; $+59$ for $N(CH_3)_3^+$ |

**3.3   Trace gas measurements**

Carbon monoxide (CO) was measured by the Aerolaser ultra-fast CO monitor model AL5002 which is based on VUV-fluorimetry (Gerbig et al., 1999; Scharffe et al., 2012). The sensor makes use of the excitation of CO at 150 nm. Therefore, UV





radiation is emitted by a resonance lamp excited by Radio Frequency discharge. An optical filter consisting of two $CaF_2$ lenses narrows the wavelength band of the emitted UV radiation to 150 nm. The fluorescence is captured at a right angle by means of a photomultiplier tube (PMT) with suprasil optics. The instrument was modified to allow in-situ calibrations during in-flight operations. During measurement flights of ACLOUD, regular calibrations were performed on 15 min to 30 min time intervals

using a NIST traceable calibration gas with a known CO concentration at atmospheric levels. Each calibration was followed by a zero measurement. This calibration procedure was used to determine and correct instrumental drifts. The precision of the calculated CO mixing ratio for ACLOUD is 1.5 ppbv. The remaining temporal stability of the CO concentration which is mainly affected by temperature variations and estimated with 4 ppbv. These values result in a total uncertainty for CO of 4.5 ppbv for all ACLOUD flights. Due to an instrumental failure on the 25 June 2017 no CO data are available for Flight #23.

For Flight #24, the CO data are only available in a reduced time resolution.

Concentrations of carbon dioxide ($CO_2$) and water vapor ($H_2O$) were measured by the LI-7200 closed $CO_2$/$H_2O$ Analyzer from LI-COR Biosciences GmbH (Burba et al., 2010; Lampert et al., 2018). The simultaneous measurement of these two gases accounts for $CO_2$-$H_2O$-interference corrections. Infrared light emitted by an optical source passes a chopper filter wheel and then enters the sample path. Behind the sample path a temperature controlled lead selenide detector measures the remaining

intensity from which the absorption is derived. The absorption ratio of $CO_2$ and $H_2O$ in the sample path was then used to calculate the density and thus the mixing ratio of both gases. The LI-7200 instrument was mounted in a 19", 3HE rack mount including additional components for flow control and in-situ-calibrations during in-flight operations. Similar to CO, calibrations were performed in time intervals of 15 min to 30 min using a NIST traceable calibration with a known carbon dioxide mixing ratio at atmospheric levels and water vapor close to zero. For ACLOUD, the precision of the instrument is given as 0.05 ppmv

for $CO_2$ and 3.7 ppmv for $H_2O$. The temporal stability was calculated from the mean instrumental drift and was estimated with 0.39 ppmv for $CO_2$ and 26.4 ppmv for $H_2O$. Hence, the total uncertainty for $CO_2$ and $H_2O$ amounts to 0.40 ppmv, and 26.7 ppmv, respectively.

Ozone ($O_3$) was measured by the 2B Technologies Dual Beam Ozone Monitor 205. The measurement principle is based on the attenuation of ultra-violet radiation (254 nm) due to $O_3$ absorption. The UV light passes two separate 15 cm long absorption

cells which are flushed alternately with ozone-filtered and ozone-unfiltered air. By measurement of the respective intensities the ozone mixing ratios were derived. The total uncertainty of the ozone mixing ratios for ACLOUD is determined by instrumental precision and amounts to 1.21 ppbv. The time resolution for the $O_3$ instrument is 0.5 Hz whereas all other gaseous tracers are measured with 1 Hz resolution. For Flight #14 on the 8 June 2017, ozone data are only available from take-off to 12:58:36 UTC due to a failure of the data acquisition. All trace gas measurements were merged and published in a combined data set (Eppers

et al., 2019, https://doi.pangaea.de/10.1594/PANGAEA.901209).



### 3.4 Inlets

#### 3.4.1 Counterflow Virtual Impactor

Cloud particle residues (CPR) are the dry particles that remain after the evaporation or sublimation of cloud droplets or ice particles, respectively. They are closely related to the cloud condensation nuclei (CCN) and ice nucleating particles (INP) that

form the clouds. Thus, their microphysical and chemical characterization provides important information about the aerosol properties, sources, transportation pathways of atmospheric particles that formed clouds in the atmosphere.

To identify sources and transportation pathways of cloud condensation nuclei (CCN) and ice nucleating particles (INP), the microphysical and chemical properties of cloud particle residues (CPR) were characterized by the aerosol instrumentation presented in Section 3.2. For that purpose, the CPR were sampled and distributed to the individual instruments. During ACLOUD,

a counterflow virtual impactor (CVI) was applied, which on the one hand collects exclusively non-precipitating cloud particles (droplets, ice particles) inside the cloud and on the other hand releases their residual particles for aerosol analysis (Ogren et al., 1985; Twohy et al., 2003). The cloud particle collection is achieved by blowing a so-called counterflow out of the CVI inlet tip. As a consequence, interstitial gases are completely deflected from the inlet and smaller interstitial particles that are not activated to cloud droplets or did not nucleate ice particles are considerably decelerated, stopped and blown out of the inlet. Only

larger particles could overcome the counterflow and are sampled by the CVI. During ACLOUD, the clouds were dominated by liquid droplets with a rather low amount of cloud ice which was almost not detectable by the CVI. Therefore all sampled CPR can be considered to represent cloud droplet residuals (CDR).

The minimum cloud particle size that is collected by the CVI is determined by the air velocity at the inlet tip (true air speed of Polar 6) and the amount of the counterflow. Due to the rather low air speed of Polar 6, the adjustment of the counterflow

to about $2\,l\,min^{-1}$ could minimize the lower cut-off diameter only to $8\,\mu m$, which is slightly higher than reported in previous operation of the CVI inlet (Schwarzenboeck et al., 2000). Therefore, CDR could not be sampled for the complete cloud droplet population (Mertes et al., 2005). From time to time the counterflow was raised to $12\,l\,min^{-1}$ in order to sample only the large hydrometeors in the cloud, which increased the lower cut-off size to $22$–$24\,\mu m$. After collection, the cloud particles are virtually impacted in a sampling line with a warm, dry, and particle-free carrier air. By evaporation of the liquid water and/or ice into

the gas phase, the CDR become released and are distributed to the different aerosol sensors.

To calculate concentrations of the CDR with respect to ambient cloud particle concentrations, the enrichment of the CVI needs to be considered. The enrichment factor is specified by the ratio of the air volume flows in front and within the CVI tip and can be expressed by a velocity ratio. The first velocity is identical to the true air speed of the Polar 6 and the second is calculated by the total sample flow. At typical in-cloud sampling conditions when all aerosol sensors were connected to the

CVI (except the PSAP) the CVI enrichment was around a factor of 4.5. All particle concentrations measured behind the CVI were corrected accordingly. This has the positive effect to reduces the detection limits of the instruments.

It needs to be considered, that the operation of the CVI is designed for particles entering parallel to the inlet. In case of a significant angle of sideslip (orientation of wind vector with respect to aircraft heading) not all droplets with diameters above the CVI lower cut-off size can be sampled. In that case, many droplets move on particle trajectories that have larger





deviation angles with respect to the CVI inlet tip and are thus not collected. The extend of this effect, which was quantified by the aspiration efficiency, was inferred from the size resolved cloud droplet number concentration measured by the cloud particle probes that are sensitive down to diameters of $5\,\mu m$ (mainly CDP and SID-3, cf. Section 3.1). For the ACLOUD measurement, the aspiration efficiency was estimated to vary between 0.2 and 0.8. During the first half of ACLOUD (Flights #7-5    15), measurements by SID-3 were used to calculate the aspiration efficiency, while in the second half, combined measurements by SID-3 and CDP did provide a more accurate estimate. For Flights #8 and #10, when neither SID-3 nor CDP were measuring, the aspiration efficiency of the previous flight was applied.

To convert the in-cabin measurements to ambient CDR properties, the CVI sampling efficiency needs to be characterized. It quantifies the ratio between number concentration of the sampled CDR and total number of cloud particles detected by the
SID-3 or CDP probe. The sampling efficiency is affected by the aspiration efficiency and depends on the shape of the cloud droplet size distribution, which can change within the cloud profile. In lower cloud levels which are typically dominated by small cloud droplets (smaller than CVI cut-off diameter), the sampling efficiency is lower than in the upper cloud parts, where most of the cloud particles are larger than the CVI cut-off size. Thus, in the upper cloud layers the sampling efficiency was almost identical to the aspiration efficiency. Assuming that there were no differences between the CDR of sampled droplets
and those of droplets larger than the CVI cut-off size, the derived sampling and aspiration efficiency were used to calculate ambient residual mass concentrations.

### 3.4.2 Aerosol inlet

The standard aerosol inlet on Polar 6 is a stainless steel inlet (Leaitch et al., 2016; Burkart et al., 2017) mounted on the front top of the aircraft, ahead of the engines. The inlet tip is a shrouded diffuser ($0.35\,cm$ diameter at intake point). Inside the
cabin, the inlet was connected to a $1.9\,cm$ stainless steel manifold of which sample lines were drawn to the various instrument racks using angled inserts. The manifold exhaust flowed freely into the back of the cabin such that the intake flow varied with aircraft true airspeed. Due to the rather low flight speed, the manifold was not significantly over-pressured. For a true airspeed of $90\,m\,s^{-1}$, the total flow at the intake point was approximately $55\,l\,min^{-1}$, based on the sum of flows drawn by the instrumentation and bypass ($13\,l\,min^{-1}$) and the measured exhaust flow into the cabin ($42\,l\,min^{-1}$). Sampling speed in the
inlet tip was approximately isokinetic for the airspeeds during ALOUD, such that the particle transmission by the inlet was near unity for particles from $20\,nm$ to about $1\,\mu m$.

### 3.4.3 Gas inlet

Two different inlets for trace gases were operated on Polar 6 (Leaitch et al., 2016). CO and $O_3$ were sampled through an inlet designed with a Teflon tube of $0.40\,cm$ outer diameter (OD). The air was passively pushed into the inlet by the aircraft forward
motion in combination with a rear-facing exhaust Teflon line ($0.95\,cm$ OD) that reduced the line pressure. The sample flow was continuously recorded and remained almost stable at approximately $19\,l\,min^{-1}$. For the sampling of $CO_2$ and $H_2O$, a separate gas inlet was used to avoid interaction of water vapor with the walls of the tubing. Therefore, this inlet is made of a stainless

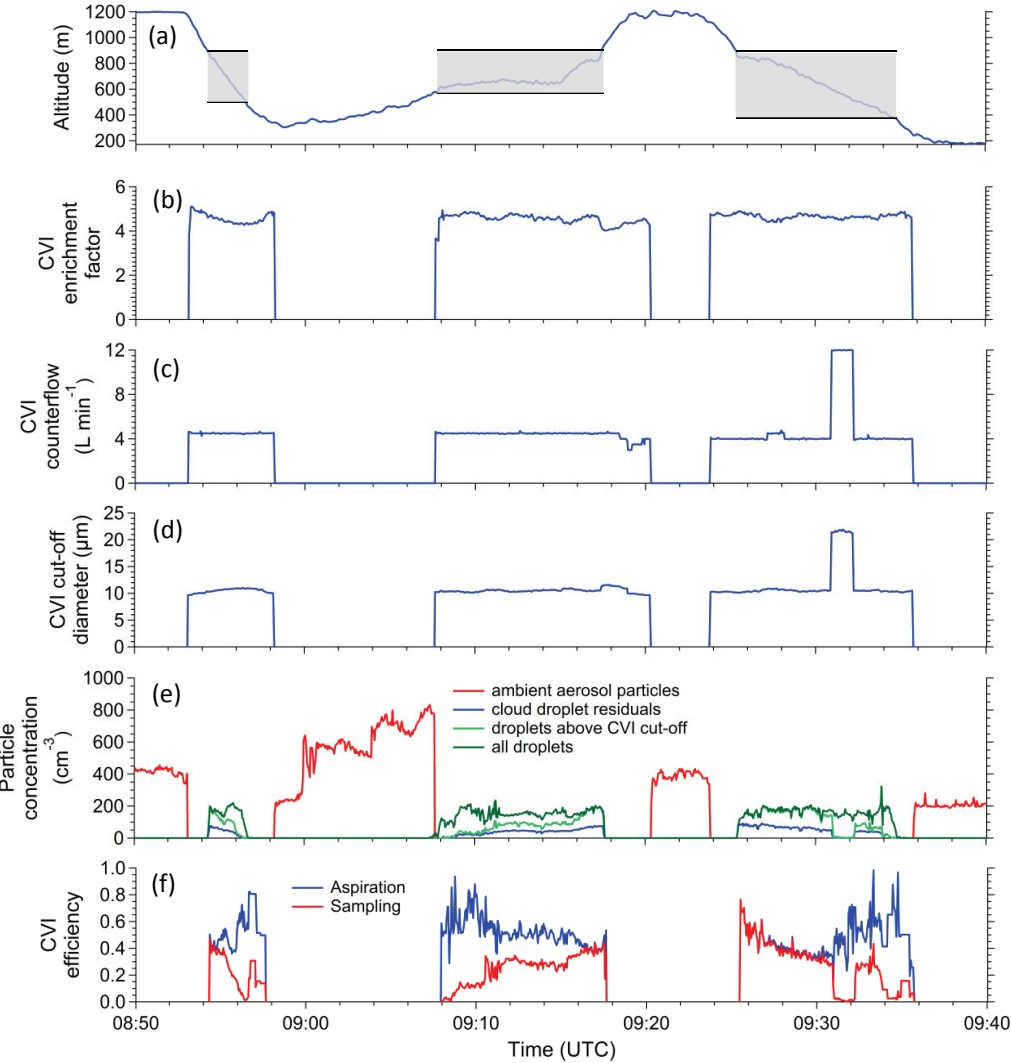

**Figure 5.** Time series of the flight altitude indicating the in-cloud flight sections (a) the CVI enrichment factor (b), counterflow (c), cut-off diameter (d), CDR, ambient particle and droplet concentration (e) and CVI aspiration and sampling efficiency (f) for Flight #11 on 2 June 2017.

steel tube (0.40 cm OD). Similar to the Teflon inlet, the air flow was passively induced by the aircraft motion. For the typical true air speeds of Polar 6 flown during ACLOUD, a continuously flow of approximately $17 \, l \, min^{-1}$ was obtained.



### 3.4.4 Operation of CVI and aerosol inlet

The parallel operation of the aerosol and CVI inlet aims to characterize both ambient aerosol particles and CDR. Therefore, most aerosol instruments were connected to both inlets allowing to switch between the inlets during flight. Table 4 summarized the configuration operated during ACLOUD. Unfortunately, the de-icing of the aerosol inlet did not always work properly. Flying in clouds with super-cooled liquid droplets, the inlet occasionally froze up. During these times, the aerosol inlet was clogged, ambient aerosol particles were sampled through the CVI inlet operating without counterflow. To avoid the risk of loosing data due to icing, the strategy of the inlet operation was changed during the campaign, connecting all instruments permanently through the CVI by switching off the counterflow, when Polar 6 was clearly out of clouds (see Table 4).

**Table 4.** Configuration of aerosol instruments and the inlet systems. Flight numbers indicate weather the instrument was switching during a flight between aerosol and CVI inlet or weather it remained connected to the CVI measuring ambient aerosol by switching off the CVI counterflow.

| Instrument | Aerosol & CVI inlet | CVI inlet only |
|---|---|---|
| UHSAS-1, CPC, PSAP | – | #4–25 |
| UHSAS-2, OPC, SP2,ALABAMA | #7, #8, #9, #10, #12, #15, #19, #20, #22, #23 | #11, #13, #14, #16, #17, #18, #21, #24, #25 |

The operation of the CVI is illustrated in Figure 5 exemplary for Flight #11 (2 June 2017) by the CVI technical parameter (enrichment factor, cut-off diameter, sampling and aspiration efficiency) and measured particle concentrations. The time series includes two descents (first and third cloud measurements) and one ascent through a cloud layer. In between, four legs in ambient conditions (two above and two below) were flown. The common procedure was to switch on/off the counterflow well before entering a cloud and well after leaving the cloud. The short outside cloud measurements were used to check the correct CVI operation indicated by zero CDR concentration measured behind the CVI (Figure 5e). As soon as the counterflow was off, the CVI inlet was operated as second aerosol inlet measuring the ambient aerosol particles. In this sampling mode no enrichment exists, but the aspiration/sampling efficiency are assumed to be 1, which was confirmed by comparison measurements at the standard aerosol inlet. Inside clouds, the CVI enrichment factor (between 4 and 5), the CVI counterflow (around $4 \, l \, min^{-1}$), and the CVI cut-off diameter (around $11 \, \mu m$) did not significantly change over the whole flight, except for a short period at about 09:33 UTC. For this leg, the counterflow was substantially increased to obtain a higher cut-off diameter of about $22 \, \mu m$ and analyze larger cloud particles. Consequently, the CDR concentration dropped to almost zero what indicates that only a small number of large particles were present.

To interpret the CVI sampling and aspiration efficiency, Figure 5e shows the total cloud droplet concentration measured by SID-3. Additionally, the concentration of cloud droplets larger than the CVI cut-off diameter was calculated from the SID-3 measurements. For the cloud shown here, the cloud top is dominated by large droplets while at cloud base small droplets are in majority. Accordingly, the CVI aspiration and sampling efficiency are rather equal at cloud top. Towards cloud base, the sampling efficiency becomes smaller, while the aspiration efficiency remains rather constant.

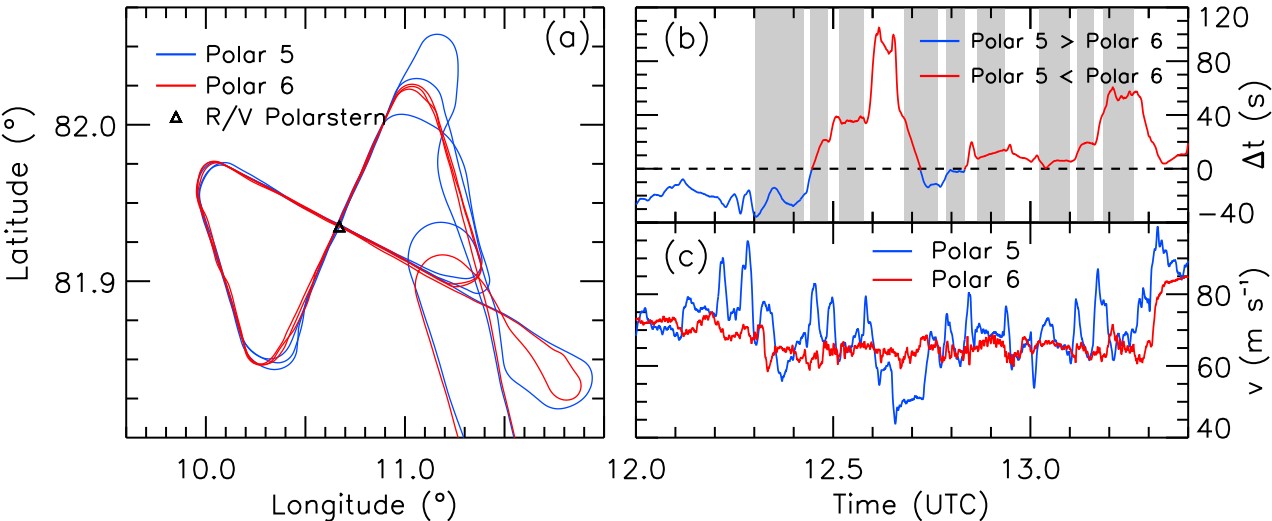

**Figure 6.** Double triangle flight track of Polar 5 and 6 on 5 June 2017 (Flight #13) close to R/V Polarstern (a). Panel (b) shows the time difference between both aircraft along the flight path. For positive values (red) Polar 5 was ahead of Polar 6 and vice versa for negative values (blue). Gray shades areas indicate the straight flight legs of the double triangle where both aircraft were coordinated. Panel c shows the flight velocity of both aircraft.

During Flight #18 (16 June 2017, 08:04 UTC), the CVI inlet heating broke and could not be repaired. In the following flights, this occasionally led the CVI inlet to freeze up when flying inside clouds. However, out-side clouds the inlet could always be de-iced so that the majority of CDR measurements and all ambient aerosol particle measurements are valid. Measurements identified to be affected by inlet freezing were removed from the data sets of the connected aerosol instruments.

## 4 Coordinated flights and intercomparison

### 4.1 Combined Polar 5 and 6 flights

The identical flight performance of Polar 5 and 6 was used to coordinate the flight patterns of both aircraft in a way that measured data can be collocated or merged into a combined data set. Collocated flights aim to combine remote sensing and in situ observations. Similar flight patterns of Polar 5 and 6 at different locations were used to extend the data set of identical instruments installed on both aircraft.

### 4.1.1 Collocated remote sensing and in situ observations

Six flights have been performed with Polar 5 and 6 flying a closely collocated flight track in different altitudes to characterize clouds (see Table 1). While Polar 5 maintained at a high flight altitude of about 3000 m for the remote sensing of cloud

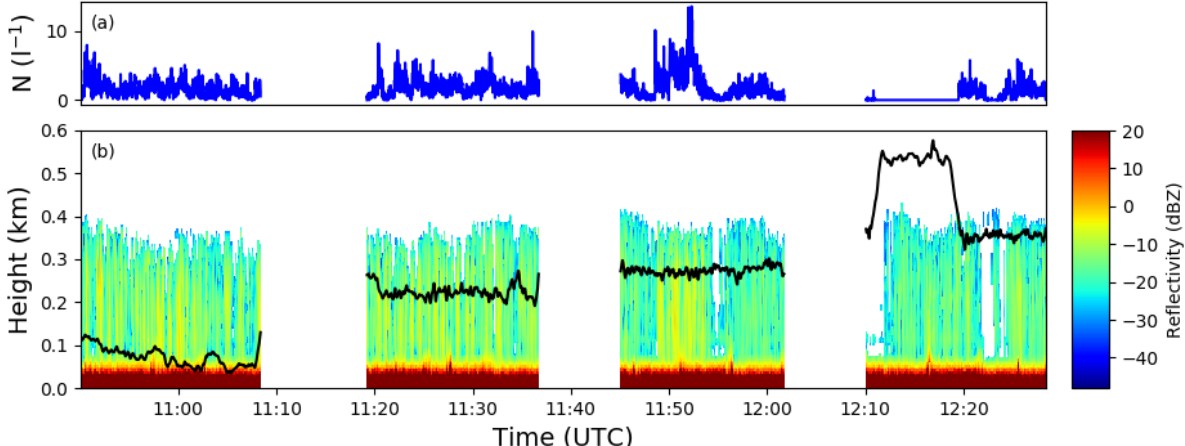

**Figure 7.** Time series of radar reflectivity profiles (b) measured on 2 June 2017 (Flight #13) during a double-triangle flight pattern (compare  6). The flight altitude of Polar 6 operating in the cloud layer is indicated by the black line. Ice crystal number concentration of particle larger than 125 μm measured by the CIP instrument along this flight track are shown in (a). The data gaps result from extended turns of the aircraft when both aircraft were not well collocated and data was removed from the comparison.

properties, Polar 6 remained in, below, or little above the cloud layer measuring cloud and aerosol particle properties in situ. The close collocation allows to analyze the same clouds by observations from both aircraft. Figure 6 shows an example of a double-triangle flight pattern flown on 5 June 2017 (Flight #13) close to the research vessel Polarstern. Along the two long straight legs of the double-triangle and the western short leg, both aircraft aimed to be horizontally collocated. To follow the same track with two aircraft is not difficult with modern navigation equipment. The task was to be at the same location within a short time difference to avoid changes of the cloud properties between remote sensing and in situ observations. Therefore, Polar 5 adjusted the flight speed as shown in Figure 6c and if needed, extended turns to reduce the distance between both aircraft. Figure 6b shows the time lag between Polar 5 and 6 along the flight track for the entire double-triangle pattern. When values are positive (red) Polar 5 was ahead of Polar 6 and vice verse for negative values (blue). Gray shaded areas indicate the straight flight legs of the double triangle where both aircraft tracks were coordinated. During these legs, the time difference was mostly below 40 s. Only for the last leg the separation exceeded 50 s.

Figure 7 shows a comparison of collocated remote sensing measurements obtained by MiRAC on Polar 5 and cloud in situ observations by the CIP on Polar 6. The data was obtained during a coordinated double-triangle pattern flown on 2 June 2017 (Flight #11). Extended turns, when both aircraft were not well collocated, were excluded from the analysis. The radar reflectivity (Figure 7b) shows a typical structure of Arctic mixed-phase boundary layer clouds with periodically occurring cloud rolls characterized by an enhanced radar reflectivity that is caused by the presence of ice crystals. Within the same cloud, Polar 6 was measuring at different altitudes indicated by the flight altitude in Figure 7b (black line). The ice crystal number concentration for particles larger than 125 μm, measured along this flight track by the CIP, is given in panel a. The concentration significantly

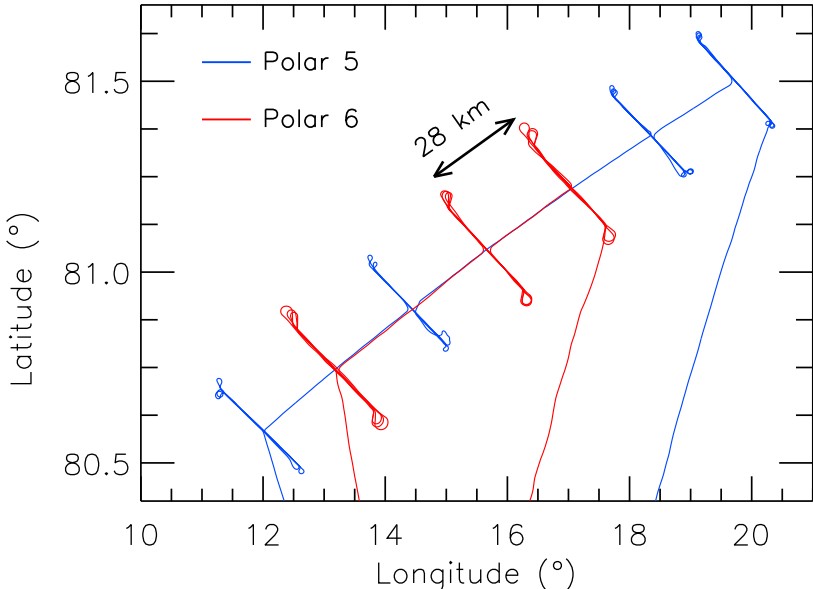

**Figure 8.** Flight track of Polar 5 and 6 on 25 June 2017 (Flight #23) measuring turbulent fluxes at different location spread over a distance of 170 km.

varies between zero and $10\,\mathrm{m}^{-3}$. These temporal (spatial) variations are clearly correlated with the changes of radar reflectivity. E.g., the cloud roll structure identified by the radar in the first and second leg (10:50-11:10 UTC and 11:20-11:35 UTC) is well captured by the variation of the ice crystal concentration measured by the CIP. Similarly, enhanced ice crystal concentrations were observed by both instruments for a longer period (larger cloud part) at around 11:50 UTC. These collocated remote

5    sensing and in situ observations are of high value for further analysis aiming to validate the remote sensing methods and to characterize microphysical processes in Arctic mixed-phase clouds.

### 4.1.2   Series of staples

The combination of both aircraft allowed for measuring vertical profiles of turbulent fluxes at a number of different locations along the main wind direction. Such flight patterns have been flown on 14, 20, and 25 June 2017 (Flights #17, #21, #23). As

10    an example the flight track of 25 June 2017 is illustrated in Figure 8. Compared to a single aircraft mission, the number of profiles was increased by a factor of two. The six profiles (six legs with each 30 km length) extend over a distance of 170 km with 28 km separation of the individual profiles. This high spatial resolution of profiles was chosen to understand the degree of spatial variability of turbulence over the marginal sea ice zone. However, it needs to be assured that measurements of both instruments on both aircraft can be merged into a single data set.





## 4.2 Merged Polar 5 and Polar 6 data

Data sets of identical instruments operated on both Polar 5 and Polar 6 can be merged to extend the scientific data analysis. Therefore, the data needs to agree within specific uncertainty ranges. To test the agreement, a coordinated flight with Polar 5 and 6 flying in close distance of about 100 m was performed on 9 June 2017 (Flight #15). The coordinated flight formation was

remained for one hour of flight time including a joint ascent and descent. Between about 1500-3100 m altitude, a cloud layer was present. Examples of the wind vector, air temperature, and broadband radiation during the comparison flight are presented in the following.

### 4.2.1 Horizontal wind vector

The horizontal wind vectors measured by Polar 5 and Polar 6 are shown in Figure 9. The $u$ and $v$ wind velocity components

are presented as vertical profiles separated into measurements during a subsequent ascent (panel c, d) and descent (panel a, b). The horizontal distance between both aircraft was roughly 100 m and the vertical distance typically 10 m.

For both wind components, the profiles measured on Polar 5 and 6 are in close agreement within $\pm 1\,\mathrm{m\,s^{-1}}$ and are both able to reproduce even very small-scale variability down to vertical scales of about 20 m. Only for altitudes below 800 m of the ascent the differences between the measurements are larger, due to a larger vertical separation of both aircraft.

The agreement for both profiles, ascent and descent, indicates, that the calibrations of the nose booms properly correct the effects of the dynamic pressure which typically act differently during ascent and descent. High-frequent variability of the wind vector naturally differs due to the remaining horizontal separation of both aircraft. However, the measurements in the more turbulent cloud layer above 2200 m illustrate, that the magnitude of the fluctuations is well captured by both nosebooms which is important for the calculation of turbulent fluxes. A similar quantitative agreement is obtained for the vertical velocity

measured by both aircraft (not shown).

### 4.2.2 Air temperature and humidity

Figure 10a and b shows time series of air temperature and relative humidity (over water) measured on Polar 5 and Polar 6 during the collocated flight section on 9 June 2017 (Flight #15). The correlations between the instruments during this section are illustrated in Figure 11c and d. The flight section includes an ascent and descend and, therefore, covers a significant range

of atmospheric conditions with temperatures between -7 °C and 4 °C and relative humidity of 45-95 %.

For the entire time series, the Pt100 of Polar 5 shows slightly lower temperatures of about 0.2 K below the measurements on Polar 6. However, the small scale variability is reproduced by both aircraft indicated by the Pearson's correlation coefficient $r = 0.998$ close to 1.0. Only in the inversion layer (08:55 UTC), characterized by the fast increase of temperature with height, larger differences were observed which are likely caused by a slight vertical distance between both aircraft.

The humidity sensors also capture the atmospheric structures in very fine detail ($r = 0.948$). However, a significant bias was observed between Polar 5 and 6 with higher humidity measured by Polar 5. On average, the bias is about 5 % relative humidity but it obviously changes with time (little differences in the end of the flight section). These differences have to be taken into



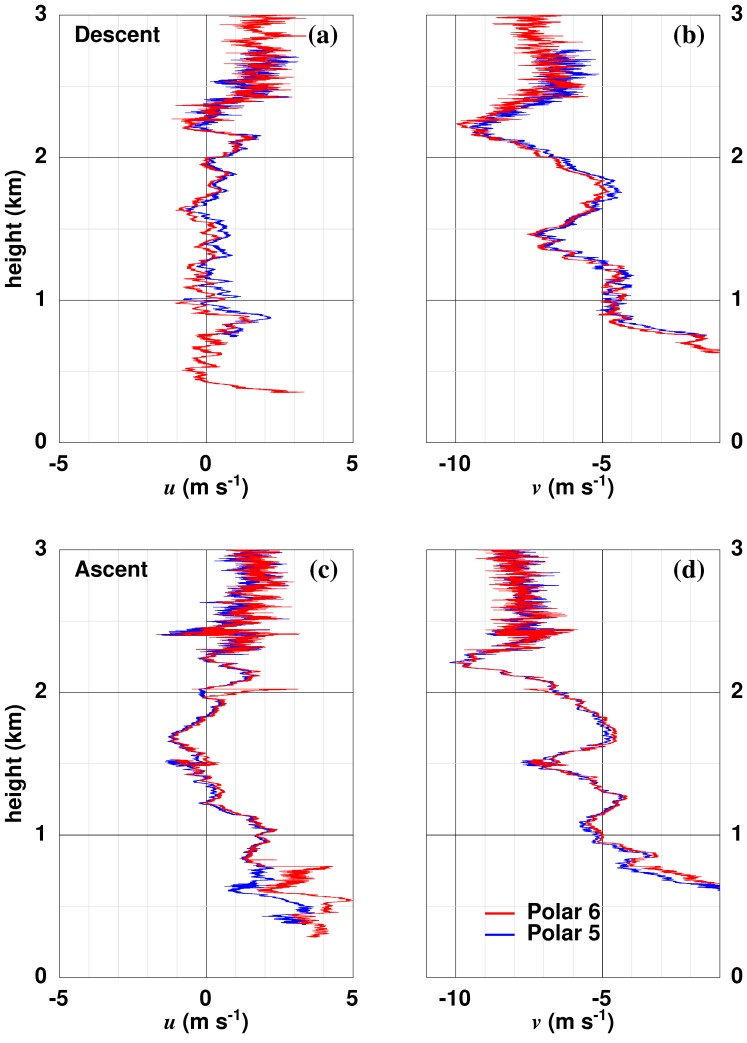

**Figure 9.** Vertical profiles of the horizontal wind components $u$ and $v$ measured by Polar 5 and Polar 6 during the close formation comparison flight on 9 June 2017. While panel a and b show the profiles obtained during a descent, data from the following ascent are given in panel c and d. The horizontal distance between both aircraft was roughly 100 m and the vertical distance typically 10 m.

account when analyzing microphysical properties within clouds where small differences of relative humidity may affect the formation of cloud particles. For this purpose, instruments measuring the absolute humidity such as the LiCOR integrated in the nose boom of Polar 5 need to be applied.

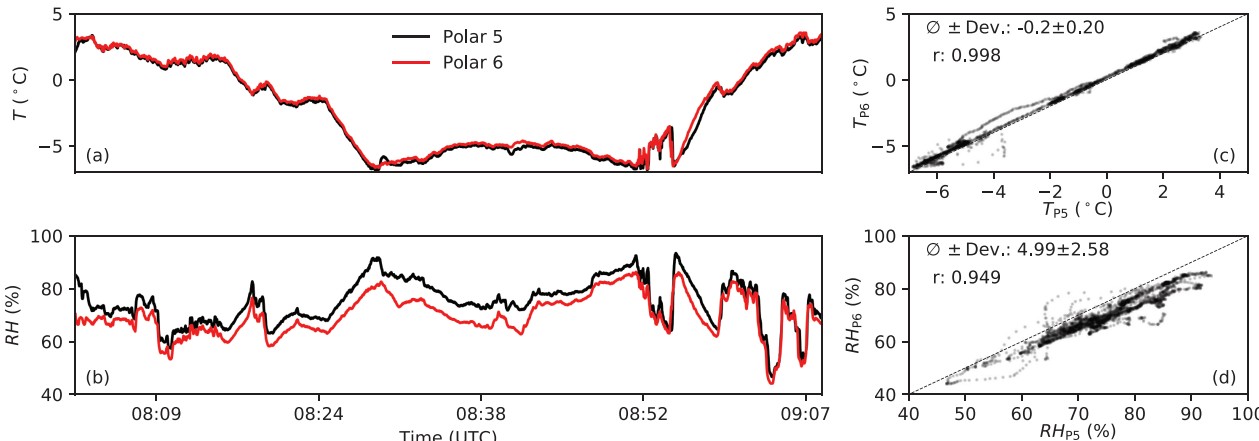

**Figure 10.** Time series of air temperature $T$ (panel a) and relative humidity $RH$ (panel b) measured on Polar 5 (P5) and Polar 6 (P6) during a collocated flight section on 9 June 2017 (Flight #15). Panel (c) and (d) show the scatter plot of Polar 5 versus Polar 6 measurements for both quantities. $\varnothing$ gives the mean and "Dev" the standard deviation of the difference of $T$ and $RH$ measured on Polar 5 and 6. $r$ denotes the Pearson's correlation coefficient.

### 4.2.3 Broadband radiation

For the coordinated section of Flight #15 (9 June 2017), Figure 11a-d shows time series of all four components of the radiative energy budget, up- and downward irradiance for the solar and terrestrial spectral range. The correlations between the Polar 5 and 6 time series are given in Figure 11e-g. The time series includes periods when stratiform clouds were present above the aircraft

(8:14–8:31 UTC and after 8:55 UTC) and conditions with cloud-free sky (8:31-8:55 UTC). Before 8:14 UTC, occasionally cirrus has been in front of the Sun. The downward solar and terrestrial irradiance, $F_{\mathrm{solar}}^{\downarrow}$ and $F_{\mathrm{terr}}^{\downarrow}$ agree well for both regimes; low $F_{\mathrm{solar}}^{\downarrow}$ and high $F_{\mathrm{terr}}^{\downarrow}$ in cloudy situations and high $F_{\mathrm{solar}}^{\downarrow}$ and low $F_{\mathrm{terr}}^{\downarrow}$ in cloud-free situations. Differences occur when horizontally inhomogeneous clouds have been above the aircraft (8:11 UTC and 8:52 UTC), or during the ascent and descent through the mid-level cloud (8:30 UTC and 8:55 UTC). In these cases, the small horizontal displacement of both aircraft is

sufficient to measure different parts of the cloud and radiation field, and explains the enhanced differences of $F_{\mathrm{solar}}^{\downarrow}$ and $F_{\mathrm{terr}}^{\downarrow}$ in the intermediate range of irradiances between cloud-free and cloudy measurements. However, the mean deviation is below $1\,\mathrm{W\,m^{-2}}$ for all quantities.

## 5 Data availability

All data listed and described here are published in the World Data Center PANGAEA (Ehrlich et al., 2019b, https://doi.org/

10.1594/PANGAEA.902603). Table 5 links each instrument to individual data sets and references. Within PANGAEA, these data are tagged with "ACLOUD" (https://www.pangaea.de/?q=keyword:"ACLOUD"), and "AC3" (https://www.pangaea.de/
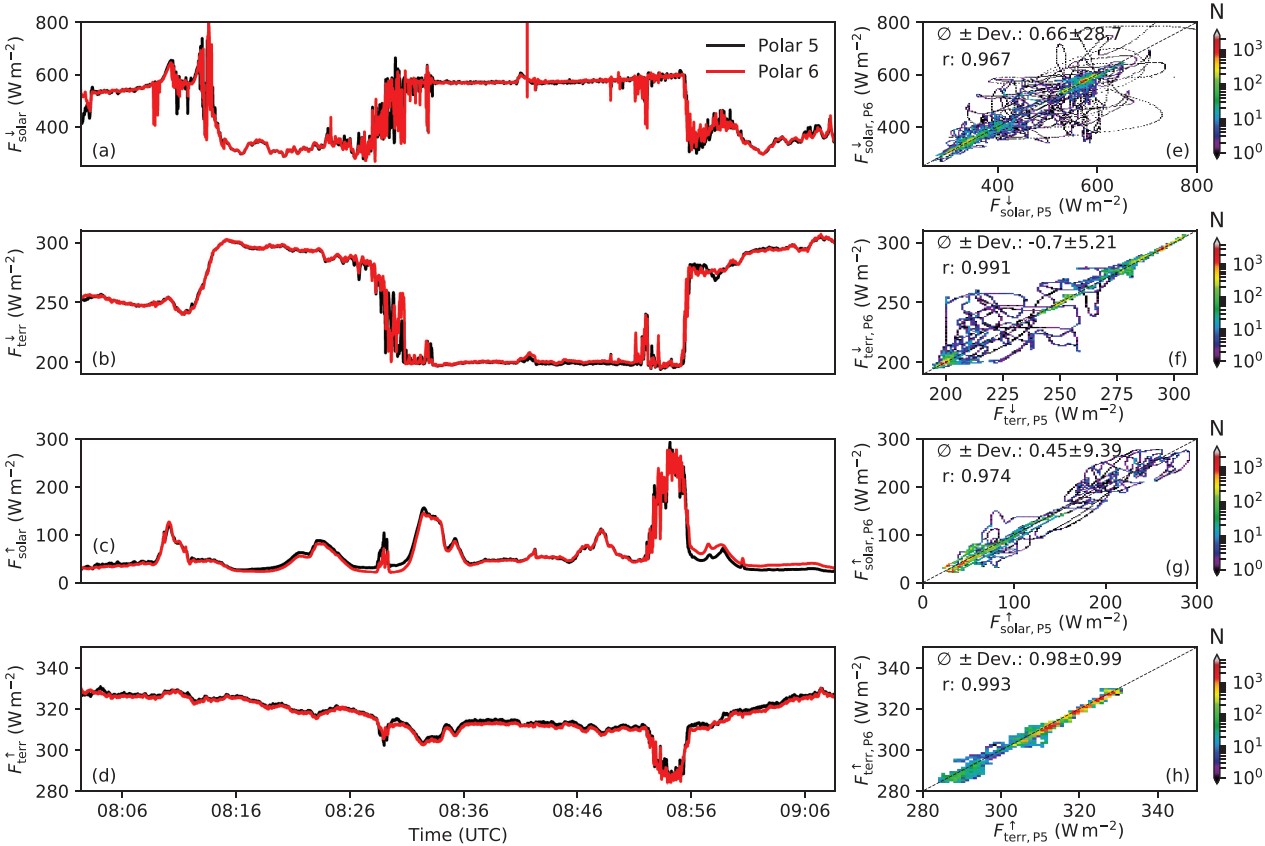

**Figure 11.** Time series of up- and downward solar irradiance (panel a, c) and up- and downward terrestrial irradiance (panel b, d) measured on Polar 5 (P5) and Polar 6 (P5) during a collocated flight section on 9 June 2017 (Flight #15). Panel e-f show the scatter plot of Polar 5 versus Polar 6 measurements for all four irradiances. The color code indicates the number $N$ of data points for each combination of values. $\varnothing$ gives the mean and "Dev" the standard deviation of the difference of irradianes measured on Polar 5 and 6. $r$ denotes the Pearson's correlation coefficient.

?q=project:label:AC3) referring to the aircraft campaign and the overarching project (AC)³. Within (AC)³, other accompanying data such as long term observations in Ny Ålesund and measurements during the Polarstern cruise PASCAL are published in PANGAEA.

The data availability and quality of each data set are indicated in Table 6 by a color code. Green indicates a complete and valid data set. Partly incomplete or defective data which allow a limited analysis are labelled yellow. Red indicates completely missing data. Empty boxes show flights when the instrument was not operated (e.g., flight without clouds). Detailed information on the data quality are given in the meta data of each data set.



## 6 Conclusions

The ACLOUD campaign provides a comprehensive in situ and remote sensing observational data set characterizing the Arctic boundary layer and mid-level cloud. All data are published in the PANGAEA data base by instrument-separated data subsets. This paper aims to give an overview of the instrument specification, data processing, and data quality. For detailed information, references are provided. It was highlighted, how the scientific analysis of the ACLOUD data benefits from the operation of two identical aircraft. True collocated data of in situ and remote sensing observations have the potential to validate remote sensing methods, e.g., identify their sensitivities with respect to ice particles. Merging the data of identical instruments operated on both aircraft extends the spatial coverage of atmospheric quantities and turbulent and radiative energy flux measurements. The different cloud remote sensing techniques operated on Polar 5 can be combined to explore the synergy of multi-instrument cloud retrieval.

**Table 5.** Overview of data sources in PANGAEA for all individual data sets of ACLOUD separated into Polar 5 and 6.

| | Instrument | Reference | Link to data source in PANGAEA |
|---|---|---|---|
| **Polar 5** | Master tracks | (Ehrlich et al., 2018a) | https://doi.org/10.1594/PANGAEA.888173 |
| | Noseboom meterologial data (100 Hz) | (Hartmann et al., 2019) | https://doi.pangaea.de/10.1594/PANGAEA.900880 |
| | Broadband Radiation and KT-19 | (Stapf et al., 2019) | https://doi.org/10.1594/PANGAEA.900442 |
| | Dropsondes | (Ehrlich et al., 2019a) | https://doi.org/10.1594/PANGAEA.900204 |
| | SMART | (Jäkel et al., 2019) | https://doi.org/10.1594/PANGAEA.899177 |
| | Eagle/Hawk | (Ruiz-Donoso et al., 2019) | https://doi.pangaea.de/10.1594/PANGAEA.902150 |
| | MiRAC | (Kliesch and Mech, 2019) | https://doi.org/10.1594/PANGAEA.899565 |
| | AMALi (cloud top) | (Neuber et al., 2019) | https://doi.org/10.1594/PANGAEA.899962 |
| | 180° Fish-Eye Camera | (Jäkel and Ehrlich, 2019) | https://doi.org/10.1594/PANGAEA.901024 |
| **Polar 6** | Master tracks | (Ehrlich et al., 2018b) | https://doi.org/10.1594/PANGAEA.888365 |
| | Noseboom meterologial data (100 Hz) | (Hartmann et al., 2019) | https://doi.pangaea.de/10.1594/PANGAEA.900880 |
| | Broadband Radiation and KT-19 | (Stapf et al., 2019) | https://doi.org/10.1594/PANGAEA.900442 |
| | CDP, CIP and PIP | (Dupuy et al., 2019) | https://doi.pangaea.de/10.1594/PANGAEA.899074 |
| | SID-3 (particle size distribution) | (Schnaiter and Järvinen, 2019a) | https://doi.pangaea.de/10.1594/PANGAEA.900261 |
| | SID-3 (single particle data) | (Schnaiter and Järvinen, 2019b) | https://doi.pangaea.de/10.1594/PANGAEA.900380 |
| | CVI and UHSAS-1, CPC, PSAP | (Mertes et al., 2019) | https://doi.pangaea.de/10.1594/PANGAEA.900403 |
| | UHSAS-2 | (Zanatta and Herber, 2019a) | https://doi.org/10.1594/PANGAEA.900341 |
| | SP2 | (Zanatta and Herber, 2019b) | https://doi.org/10.1594/PANGAEA.899937 |
| | ALABAMA | (Eppers and Schneider, 2019b) | https://doi.pangaea.de/10.1594/PANGAEA.901047 |
| | OPC Grimm | (Eppers and Schneider, 2019a) | https://doi.pangaea.de/10.1594/PANGAEA.901149 |
| | Trace gases (CO, $O_3$, $CO_2$, $H_2O$) | (Eppers et al., 2019) | https://doi.pangaea.de/10.1594/PANGAEA.901209 |

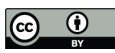



**Table 6.** Overview of data availability and quality of all core instruments shown by the color code. Green indicates a complete and valid data set. Partly incomplete or defective data which allow a limited analysis are labelled yellow. Red indicated completely missing data. Empty boxes show flights when the instrument was not operated.

| # | Date | Polar 5 | | | | | | | | Polar 6 | | | | | | | | | | |
|---|------|---------|----|----|----|----|----|----|----|---------|----|----|----|----|----|----|----|----|----|----|
| | | BD | NB | DS | SA | EH | MR | MM | AM | BD | NB | S3 | PH | CP | PP | CD | CV | AB | AR | TG |
| 4 | 23 May | | | | | | | | | | | | | | | | | | | |
| 5 | 25 May | | | | | | | | | | | | | | | | | | | |
| 6 | 27 May | | | | | | | | | | | | | | | | | | | |
| 7 | 27 May | | | | | | | | | | | | | | | | | | | |
| 8 | 29 May | | | | | | | | | | | | | | | | | | | |
| 9 | 30 May | | | | | | | | | | | | | | | | | | | |
| 10 | 31 May | | | | | | | | | | | | | | | | | | | |
| 11 | 02 June | | | | | | | | | | | | | | | | | | | |
| 12 | 04 June | | | | | | | | | | | | | | | | | | | |
| 13 | 05 June | | | | | | | | | | | | | | | | | | | |
| 14 | 08 June | | | | | | | | | | | | | | | | | | | |
| 15 | 09 June | | | | | | | | | | | | | | | | | | | |
| 16 | 13 June | | | | | | | | | | | | | | | | | | | |
| 17 | 14 June | | | | | | | | | | | | | | | | | | | |
| 18 | 16 June | | | | | | | | | | | | | | | | | | | |
| 19 | 17 June | | | | | | | | | | | | | | | | | | | |
| 20 | 18 June | | | | | | | | | | | | | | | | | | | |
| 21 | 20 June | | | | | | | | | | | | | | | | | | | |
| 22 | 23 June | | | | | | | | | | | | | | | | | | | |
| 23 | 25 June | | | | | | | | | | | | | | | | | | | |
| 24 | 26 June | | | | | | | | | | | | | | | | | | | |
| 25 | 26 June | | | | | | | | | | | | | | | | | | | |

BD: basis data acquisition, NB: nose boom, DS: dropsondes, SA: SMART-albedometer, EH: AISA Eagle/Hawk, MR: MiRAC Eagle/Hawk, MM: MiRAC radar, MM: MiRAC microwave radiometer, AM: AMALi, S3: SID-3, PH: PHIPS, CP: CIP, PP: PIP, CD: CDP, CV: CVI, AB: ALABAMA, AR: aerosol rack, TG: trace gases





A series of ongoing studies made already use of the ACLOUD data concentrating on some of the highlights presented by Wendisch et al. (2019). These studies are collected in the interjournal special issue of Atmospheric Chemistry and Physics and Atmospheric Measurement Techniques, Arctic Mixed-Phase Clouds as Studied during the ACLOUD/PASCAL Campaigns in the Framework of (AC)³ (www.atmos-meas-tech.net/special_issue10_971.html). However, the data set has a lot of further potential for detailed studies on cloud-aerosol interaction, satellite remote sensing comparison, validation of cloud resolving numerical models and more. Further data products that are currently in development will be added to PANGAEA in future and will be linked to the current data set within PANGAEA via the tag "ACLOUD".

In March/April 2019, most of the ACLOUD instrumentation (remote sensing instruments and part of the in situ cloud probes) was operated on Polar 5 during the Airborne measurements of radiative and turbulent FLUXes of energy and momentum in the Arctic boundary layer (AFLUX) campaign. In early spring and a late summer 2020 it is planned to repeat the coordinated operation of both Polar 5 and 6 using the ACLOUD instrument configuration during the Multidisciplinary drifting Observatory for the Study of Arctic Climate - Airborne observations in the Central Arctic (MOSAiC-ACA) campaign as part of the MOASiC expedition within the framework of the (AC)³ project. These data will extend the ACLOUD observations in different seasons and in higher latitudes of the central Arctic and, therefore, will allow a statistically solid analysis of atmosphere, cloud, aerosol, trace gas, and sea ice properties.

*Author contributions.* The general part of the manuscript was prepared by AE, CL, and MW. Contributions to the section of the individual instruments were provided by CL, JH, DC (noseboom), SC, LLK, MM (MiRAC), HB, OE (trace gases), RD, EmJ, OJ, MS (cloud probes), AH (sun photometer), EvJ, ERD (spectral solar remote sensing), JSt (broadband radiation), SM, UK, OE, HCC, FK, JSc, MZ (aerosol instrumentation), and RN (AMALi). MB summarized the data availability in PANGAEA. All authors discussed the results and contributed to the final writing of the paper.

*Competing interests.* No competing interests are present.

*Acknowledgements.* We gratefully acknowledge the funding by the Deutsche Forschungsgemeinschaft (DFG, German Research Foundation) – Project Number 268020496 – TRR 172, within the Transregional Collaborative Research Center "ArctiC Amplification: Climate Relevant Atmospheric and SurfaCe Processes, and Feedback Mechanisms (AC)³". The LaMP acknowledges the support of the Pollution in the Arctic System (PARCS) project funded by the Chantier Arctique of the Centre National de la Recherche Scientifique–Institut National des Sciences de l'Univers (CNRS-INSU). The authors are grateful to AWI for providing and operating the two aircraft during the ACLOUD campaign. We thank the crews of the Polar 5 and Polar 6 and their technicians for the excellent technical and logistical support. The generous funding of the flight hours for ACLOUD by AWI is greatly appreciated.



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
