# Peer review of "A comprehensive in situ and remote sensing data set from the Arctic CLoud Observations Using airborne measurements during polar Day (ACLOUD) campaign"

_Earth System Science Data, 2019_

## Referee Comment (RC1) · Sebastian Schmidt (Referee) · 19 Jul 2019

This manuscript provides a comprehensive description of the ACLOUD data set, with references to DOIs and publications for individual data sets and instrumentation. Pairing the data archive itself with such a data description following the ESSD journal's philosophy is an excellent way to get the best use of the data across the user community beyond the original ACLOUD team itself. Generally, the manuscript does a good job describing the data in such a way that they can be broadly used. However, I do have a few general recommendations to improve on this aspect. I also have a few sequential

comments. I did not comment on language much. Copyediting should be done. There are a few re-occurring errors (for example, the use of a comma before "that", which is very uncommon in BE/AE; also: there should always be a comma before "which" - numerous other [minor] language issues including mismatching numerus, tense etc.). While I recommended "major revisions" above, I actually think that no major changes to the content are needed. However, I volunteer (at the discretion of the editor) to re-read the manuscript after improvements proposed below have been made.

General comments:
1) The description of the individual data sets and instruments comes across as too much of a "laundry list" where the individual pieces are treated unequally. While to some extend unavoidable in this type of paper, I recommend to go through judiciously and decide what the user really needs to know, and also to add details where necessary. Would it be possible to follow one single template for the contributing data sets? I would like to point out that for the most part this is done well; CVI hits a good balance of material where measurement principles are described with sufficient detail for the user to understand its strengths (and weaknesses, where applicable). A counterexample is the lidar; for the non-expert, it is not satisfying to be confronted with channel specifics without being told about their use. In the end, the question remains whether one actually obtains extinction profiles, and if so, at what resolution (since this is not an HSRL, it is probably just backscatter). Occasionally, details are given that are not useful for the reader - why the "five-times" threshold, for example? On the other hand, the information on p11, l30-34 are useful. I would also like to note that the collective description of some instruments is nice: It is satisfying to see that a consistent cloud droplet size distribution can be derived across multiple instruments (question though: were there any bulk probes flown like a Nevzorov? It is conspicuously absent from Table 2. Why? Seems like the 1 microphysics instrument to include in the payload.)

2) To truly make the data set useful to the community, it would help to understand the motivation/genesis for the various different types of flights. After all, they were motivated by the science, not just be the objective to have P5/P6 collect some data together. It is understood that the science is described separately in other papers. However, this paper is incomplete without describing what the flights actually looked like, whether the flights delivered on their objectives etc. At the very least, list the various objectives, such as "above-cloud radiative effect", "surface cloud radiative effect", "surface characterization", "cloud microphysics profiling", "remote sensing validation", "air mass modification", "process-understand of xyz cloud type" [in no particular order]. This will allow the reader to understand the flown flight patterns, why individual maneuvers had the lengths they had etc... Is it possible to include meteorological context, and/or imagery? In other words, it is very desirable to have the philosophy of the campaign with sub-objectives flow down to the execution of the individual flights. This is, to some extent, more important than the description of the instrument, which can be found in other publications. That aspect is only very sparsely covered at the moment.

Sequential comments

Table 1:
* Should come after the general strategy for ACLOUD is introduced and include objectives for modules, for example as proposed in point 2 above (e.g., "air mass modification", ...)
* spent (caption) - replace with less colloquial term
* A-Train = NASA A-Train
* What are "staples"?
* Comment above suggest that we need a nomenclature of commonly flown modules. We often find "walls", "spirals", "parking garages" in the literature. If new terms are introduced here, explain them.

[Figure]

p3,l3: "comparability" - This is not the translation for "Vergleichbarkeit" if that was the intent. "Comparability" would be understood as our ability to compare the data sets, but not how they actually compare.

p4,l9: numerous -> many

p6,l15: insert "of the sensor" between "heating" and "by"

p6,l26: Insufficient description how eddy covariance method would be implemented.

p6,l32-34: Run-on sentence, re-write

p7,l11: 12-15 nm: Is this sampling or resolution? If resolution, what is the sampling?

p7,l21: remain -> maintain

p8,l10: convoluted -> should this be "convolved"?

p8,l14: have been –> were (multiple occurrences throughout document)

p9,l1-2: check language (punctuation, numerus)...

p9,l2: Why is the comparison limited to pitch/roll angle < 2?

p9,l7: "less than 1%" This is a bit unclear. The deviation of SMART from Eagle is 0.02 at about 0.2 in radiance units (Fig 1e). Isn't that 10%?

p9, l33: The KT-19 is not a broadband radiometer - quite the opposite. Why is it in this section?

p10,l10: off nadir (along track backwards) is a bit contradictory. Is it along track or off-nadir? If off-nadir, why was it mounted in this way (25 deg off)?

p11,l8: "which successively stem from..." This is unclear. How is the frequency related to the "center of the emission line" and (supposedly) sensed atmospheric level. Isn't this rather a matter of atmospheric opacity (regardless of where the emission line center is located), which translates to the location of the weighting function maximum (in remote sensing terms)? I am probably getting this completely wrong based on the text provided.

p12,section 2.6: should be completely re-written, too many things are unclear. A sunspot is used for measuring solar radiation and solar irradiance? First off, what is the difference - does this refer to sky radiance vs. direct-beam radiance? The use of "solar" for a wavelength range (if that is the intent) is especially confusing here. How about shortwave, visible or near-infrared? Second, why do sunspots (as in the "sunspot cycle" matter here? Or is it literally a "spot of sunlight"? If so, please don't use that term, which is historically reserved for something else. Third, how can a diaphragm do the "focusing"? If anything, it will diffuse radiation, quite the opposite of focusing.

p13,l19: Please describe the "Monte Carlo method" or cite paper.

p15,l9/10: Give some quantitative specifics on which parameters are provided for "sphericity", "shape" and "mesoscopic cystal complexity". Are all of these numbers? How can they be interpreted by the user? For example, does "shape" provide different information compared to "complexity"?

p15,section 3.1.4: What is the size range?

p16,figure 3: Does the MC correction just do the Mie correction as described, or does it also entail other standard correction as developed over the decades for single-droplet counters/sizers?

p17,l14 vs. l16: Throughout the manuscript, different values are stated for the upper size limit of the isokinetic inlet and the instruments behind it. While not contradictory, I recommend going through the paper again to make sure there's consistency.

p17: Why was no nephelometer flown? Seems standard equipment.

p17: Are all the aerosol size distributions "dry" or measured at ambient humidity (unlikely). Was f(RH) measured (unlikely if no nephelometer was flown). Why is aerosol humidification considered irrelevant for this particular campaign?

p18: How do the rBC measurements by the SP-2 fit in with the rest of the aerosol measurments?

p19,Table 3: Listing species is helpful, but doesn't per se allow attribution of aerosol type/source. How can the information from the different sensors (PSAP, neph if applicable, SP-2, mass spectrometer) be combined to retrieve broader aerosol typing? In isolation, the information provided here may not be helpful to the data user.

p19,l17: "Therefore" does not mean "Dafuer" Use "to that end" or "to achieve that" or "to do that"

p21,l16: "not detectable by the CVI" As written, this sentence suggests that under different circumstances (higher ice crystal concentrations), the CVI would be able to distinguish between crystals and droplets. But is that true?

p21,l26: "with respect to" -> "relative to"?

p21,l31: This begs the question what the detection limit of the instruments behind the counterflow impactor (or regular inlet) are...

p22,l1: "extend" -> "extent"

p22,l26: How does the transmission (=sampling efficiency?) fall off after 1 micron?

p24,l7: "loosing data" -> "losing data"

p24,l1: "aims to characterize" -> "aims at..."

p24,l9: "exemplary" does not mean "beispielsweise" - it means "outstandingly good" (or sometimes "serving as a deterrent/bad example") - please revise unless you mean one of the two.

p24,l25: "rather" -> "more or less" ?

p25,l2: "out-side" -> "outside"

p25, section 4: This would be the opportunity to give some examples how the P5/P6 together achieve the general goals of the campaign, but this is only done in terms of instrument synergies, and not in terms of the fulfillment of mission requirements/goals. Of course, that would only be possible if those were stated at the beginning. I strongly recommend that the authors consider adding such a description as proposed in the general comments. The manuscript would benefit tremendously form that addition (IMO).

p27,Figure 8: There are not enough details provided. How about adding imagery for context, and a cross section of one of the short "fishbone" segments to allow the reader to see the vertical structure? How does the length of the short/long legs optimize the sampling / how does it fulfill mission requirements? This is very useful information for the reader.

---

## Referee Comment (RC2) · Anonymous Referee #2 · 19 Aug 2019

General review comments: This manuscript introduces the in situ and remote sensing data set from the Arctic CLoud Observations Using airborne measurements during polar Day (ACLOUD) mission. Mission concept, vehicles, instruments and data obtained are comprehensively documented. Since this is simply a documentation for a measurement mission, this reviewer cannot find apparent flaws in the paper, thus recommend it to be published.

Minor corrections:

[Figure]

1. The text should be more concise. The authors should find way to significantly reduce the length.

2. Section 2.3

"Upward and downward broadband irradiances were measured by pairs of CMP 22 pyranometers and CGR4 pyrgeometers, covering the solar (0.2-3.6 $\mu$m) and thermal-infrared (4.5-42 $\mu$m) wavelength range, respectively. Both aircraft, Polar 5 and 6, were configured with an identical set of instruments and sampled with a frequency of 20Hz. In stationary operation, the uncertainty of the sensors is less than 3% as characterized by the calibration of the manufacturer and evaluated by, e.g., Gröbner et al. (2014). For the airborne operation of the fixed-mounted sensors, the misalignment of the aircraft was corrected by applying the approach by Bannehr and Schwiesow (1993), and Boers et al. (1998), which was applied for the downward direct solar irradiance. Therefore, the fraction of direct solar radiation was estimated using radiative transfer simulations (cloud free and cloud covered). The simulations were based on available in-flight observations and consider the temperature and humidity profiles and cloud cover. In case of clouds, the cloud optical thickness was fixed to a representative value of 5. The upward solar radiation as well as the upward and downward terrestrial radiation were assumed to be isotropic and were not corrected for the aircraft attitude. ..."

This paragraph shows the RT model is used in the data, but the justification and uncertainty of this treatment is not well discussed.

This paragraph also assumes "The upward solar radiation as well as the upward and downward terrestrial radiation were assumed to be isotropic". This is not valid for solar radiation. What's the effect of this assumption?

---

## Author Comment (AC1) · 29 Oct 2019

The comments of the reviewer have been helpful to improve the manuscript. The detailed replies on the reviewers comments are given below.

The reviewers comments are given in bold while our replies are written in regular roman letters. Citations from the revised manuscript are given as indented and italic text.

**Detailed Replies**

**The text should be more concise. The authors should find way to significantly reduce the length.**

We did go through the entire manuscript and reduced the text where it was possible without removing important details. However, still the manuscript did not significantly reduce in length. Due to the number of instruments (two fully equipped aircraft, 20 individual data sets) no further reduction is possible without loosing the main intention of the manuscript, which is to describe the data and data processing for new data users. We expect that most readers who are interested in the data, look probably only for a certain group of data. Thus we do not consider the manuscript length as critical. E.g.: If someone wants to use remote sensing observations, he or she only has to look into the section of Polar 5 and may skip the Polar 6 part.

**Section 2.3: This paragraph shows the RT model is used in the data, but the justification and uncertainty of this treatment is not well discussed.**

The radiative transfer simulations were not used to replace the measurements, if that is what the reviewer understood. The simulations only provide the relative number of the fraction between direct and solar irradiance, which cannot be measured on the aircraft. This fraction is used to weight the correction of the downward irradiance following the common approach by Bannehr and Schwiesow (1993). The contribution of uncertainties of the direct fraction to the downward radiance strongly depends on solar zenith angle and aircraft attitude. For $60°$ solar zenith angle, roll and pitch angle of $5°$, $5\%$ uncertainty of the direct fraction amounts to a total uncertainty of less than $1\%$.

To make this better understandable we changed the section into:

*This correction is valid only for the downward direct solar irradiance. Therefore,*
*the relative fractions of direct and diffuse solar radiation were estimated using radiative transfer simulations (cloud free and cloud covered). The simulations were updated continuously based on available in-flight observations and consider the temperature and humidity profiles and the presence or absence of clouds. For the conditions during ACLOUD, a 5 % uncertainty of the simulated fraction of direct radiation amounts to less than 1 % uncertainty of the corrected downward irradiance.*

**This paragraph also assumes "The upward solar radiation as well as the upward and downward terrestrial radiation were assumed to be isotropic". This is not valid for solar radiation. What's the effect of this assumption?**

This sentence might have been misleading. The point we wanted to make is that upward solar irradiance was not corrected for the aircraft misalignment. This is common procedure because of two reasons. First, a correction would require knowledge on the exact distribution of the radiation field, which is not measured and is difficult to estimate from simulations. Second, the upward radiation is way less anisotropic as the downward radiation (direct solar radiation) and the effects of the aircraft misalignment are little. A perfect isotropic radiation field would cause no effects at all. But it's true that our argumentation was wrong and misleading.

We rephrased this sentence to avoid any misunderstanding.

*The upward solar radiation as well as the upward and downward terrestrial radiation cannot be corrected for the aircraft attitude. However, these components are characterized by a nearly isotropic radiation field compared to the downward radiation and the effects of a misalignment is minimal for a nearly level sensor (Bucholtz et al. 2008). To limit the remaining uncertainties due to the aircraft movement, measurements with roll and pitch angles exceeding $\pm 4°$ were removed from the data set.*

---

## Author Comment (AC2) · 29 Oct 2019

The comments of the reviewer have been helpful to improve the manuscript. We are especially thankful for pointing at the missing objectives and flight patterns of the individual flights, which significantly increased the value of the manuscript for potential readers.

The detailed replies on the reviewers comments are given below. The reviewers comments are given in bold while our replies are written in regular roman letters. Citations

from the revised manuscript are given as indented and italic text.

**Detailed Replies**

**The description of the individual data sets and instruments comes across as too much of a "laundry list" where the individual pieces are treated unequally. While to some extend unavoidable in this type of paper, I recommend to go through judiciously and decide what the user really needs to know, and also to add details where necessary. Would it be possible to follow one single template for the contributing data sets?**

We agree, that the sections are not perfectly balanced. It is very challenging to have a common description of all instruments, which all have a different degree of complexity in methods and data processing. A template would have the risk to skip details which apply only for a single instrument. Already the table of instrument specifications shows, that general characteristics are hardly to find. However, we did go through all sections and tried our best to harmonize the manuscript. In this sense, the specific reviewer comments helped a lot.

**A counterexample is the lidar; for the non-expert, it is not satisfying to be confronted with channel specifics without being told about their use. In the end, the question remains whether one actually obtains extinction profiles, and if so, at what resolution (since this is not an HSRL, it is probably just backscatter). Occasionally, details are given that are not useful for the reader - why the "five-times" threshold, for example?**

AMLAi is a backscatter lidar which is stated in the beginning of the section. The application of the different lidar channels has been given in the second part of this section. However, the lidar section was rewritten and sharpened taking the reviewer remarks into consideration. E.g.: the use of the channels and the potential lidar products are
now summarized in the beginning.

> *The backscattered intensities can be converted into attenuated backscatter co-efficients, depolarisation ratio at 532 nm, and the color ratio (532 nm to 355 nm) to analyze cloud and aerosol particles*

The AMALi provides aerosol backscatter coefficients and does not provide a direct extinction measurement. Extinction profiles, in general, can be derived from the lidar observations and be used for a detailed analysis beyond the cloud top altitude (e.g. characterization of aerosol particles). However, we did not include these lidar profiles in the ACLOUD data base and limited the lidar data to the cloud top height for several reasons. The data processing of the backscatter profiles need special treatment depending on their specific application (clouds or aerosol). Therefore, the data processing is not yet finished and no final data version is available. The preliminary data was not published to avoid having different versions published in PANGAEA. To make this more clear in the manuscript, we restructured the AMALi section and removed any sentence referring to the unpublished data.

> *The published data set provides cloud top height derived from the preliminary lidar profiles. Clouds below the aircraft were identified from the attenuated backscatter coefficients in the 532 nm parallel channel. Each height bin of the profile, which exceeds the backscatter coefficients of a reference cloud free section by a factor of five, was labelled as cloud. Cloud top height was then defined as the highest altitude, which meets the above criterion for consecutive altitude bins. In the published data set (Neuber et al., 2019, https://doi.org/10.1594/PANGAEA.899962), cloud tops in close distance to the aircraft (less then 100 m below the flight level) and low clouds (below 30 m above the ground) are excluded. Profiles of attenuated backscatter coefficients and depolarisation ratios are available on request and not yet included in the*

*data set, because the processing of the backscatter profiles need special treatment depending on their specific application (clouds or aerosol).*

As the detection algorithm of the cloud top is important for the cloud top altitude data set, we kept the "five-times" threshold, but rephrased the sentences as given above. This minimum ratio of five was chosen after testing several values in order to optimize the analysis accounting for the instrument noise.

**Question though: were there any bulk probes flown like a Nevzorov? It is conspicuously absent from Table 2. Why? Seems like the 1 microphysics instrument to include in the payload.**

There was a Nevzorov probe flown on Polar 6. However, the estimation of the two sensor collection efficiencies is challenging given the presence of mixed-phase clouds. Especially for the mixed-phase clouds with low fractions of ice particles, the obtained $IWC$ were often biased. Therefore, we first decided against publishing the data to avoid any misinterpretation. However, we also see the value of the $LWC$ data alone and now added the processed $LWC$ and $TWC$ into the data base. The data processing and issues of the measurements in mixed-phase clouds are given in a new sub-section which reads:

*A standard Nevzorov heated wire probe (Korolev et al., 1998) was installed on the nose of Polar 6 to measure bulk liquid and total water content ($LWC$, $TWC$). The raw data were averaged over 1-second intervals and processed to compute the liquid water content based on the method described by (Korolev et al., 1998). For both sensors (total and liquid water), the collection efficiency is assumed to be equal to 1. The calculations require the true air speed, which was measured by the 5-hole probe installed at the noseboom of Polar 6. Uncertainties of Nevzorov probes have been discussed by, e.g., Wendisch and*

[Figure]

*Brenguier (2013) and Schwarzenboeck et al. (2009). The main uncertainty of the computed $LWC$ and $TWC$ is associated with the estimates of the dry-air output signal, which was determined manually right before and after the in-cloud segments of the flights. During the in-cloud segments, the dry-air signal is unknown and is obtained by linear interpolation of the before-and after-cloud values. The version of the Nevzorov probe installed on Polar 6 during ACLOUD requires manual balancing of the probe, which is done by an human operator during the flight. Some parts of the data could not be recovered when the balancing was not done on time by the operator. For the majority of clouds, the liquid water content values obtained from the $LWC$ sensor of the Nevzorov probe are in close agreement with estimates obtained by integrating the droplet size distribution measured by the CDP. The ice water content calculated from the difference of $TWC$ and $LWC$ is highly uncertain in mixed-phase clouds due to the small amount of cloud ice in the majority of clouds observed during the ACLOUD campaign and,therefore, not included in the data base (Chechin, 2019, https://doi.org/10.1594/PANGAEA.906658).*

**To truly make the data set useful to the community, it would help to understand the motivation/genesis for the various different types of flights. After all, they were motivated by the science, not just be the objective to have P5/P6 collect some data together. It is understood that the science is described separately in other papers. However, this paper is incomplete without describing what the flights actually looked like, whether the flights delivered on their objectives etc. At the very least, list the various objectives, such as "above-cloud radiative effect", "surface cloud radiative effect", "surface characterization", "cloud microphysics profiling", "remote sensing validation", "air mass modification", "process-understand of xyz cloud type" [in no particular order]. This will allow the reader to understand the flown flight patterns, why individual maneuvers had the lengths they had etc... Is it possible to include meteorological context, and/or**

**imagery? In other words, it is very desirable to have the philosophy of the campaign with sub-objectives flow down to the execution of the individual flights. This is, to some extent, more important than the description of the instrument, which can be found in other publications. That aspect is only very sparsely covered at the moment.**

The philosophy of the ACLOUD/PASCAL campaign with sub-objectives is intensively described and discussed in the campaign overview paper Wendisch et al. (2019). To make a link to the individual flights of the data set, we added Section 2 "Scientific targets of the research flights" in the revised manuscript. The Table 1 was extended to categorize these scientific targets. Further, we will provide all flight reports that have been compiled during the campaign in the supplementary of the paper.

**Table 1: Should come after the general strategy for ACLOUD is introduced and include objectives for modules, for example as proposed in point 2 above (e.g., "air mass modification", ...)**

Table 1 is now included in the new section 2 "Scientific targets of the research flights".

**Table 1: spent (caption) - replace with less colloquial term**

Was changed into:

> *In total, measurements were obtained in 165 flight hours distributed equally to both aircraft.*

**Table 1: A-Train = NASA A-Train**

Changed as suggested.

**Table 1: What are "staples"?**

We changed this into "vertical stacks" which should be more common and precise.

**Table 1: Comment above suggest that we need a nomenclature of commonly flown modules. We often find "walls", "spirals", "parking garages" in the literature. If new terms are introduced here, explain them.**

In the revised version the scientific targets are divided into four major categories: cloud remote sensing (CRS), in situ cloud and aerosol particle measurements (In Situ), surface fluxes (SF), and flux profiles (FP).

**p3,l3: "comparability" - This is not the translation for "Vergleichbarkeit" if that was the intent. "Comparability" would be understood as our ability to compare the data sets, but not how they actually compare.**

Changed into "consistency".

**p4,l9: numerous -> many**

Changed as suggested.

**p6,l15: insert "of the sensor" between "heating" and "by"**

It is rather the air itself than the sensor which is heating up by the dynamic pressure. To avoid misunderstandings, we added "of the air" in the sentence.

**p6,l26: Insufficient description how eddy covariance method would be implemented.**

It may be written misleading in the manuscript, but we do not provide calculated turbulent fluxes in the data base. This section aims only to indicate that the quality of the data is sufficient to apply the eddy covariance method. In the revised manuscript we changed this section and added some guidance how to use the data for calculating turbulent fluxes, especially pointing at what flight conditions are required. A detailed description of the turbulent fluxes is currently prepared for another publication. This would be beyond the scope of this manuscript.

*The achieved accuracy and temporal resolution of wind and temperature measurements are sufficient to derive turbulent fluxes of momentum and sensible heat in the atmospheric boundary layer with the eddy-covariance method (e.g., Busch, 1973). When using the 100 Hz data delivered to PANGAEA note that the calibration of the 5-hole probe is only valid for straight and level flights. The majority of measurements during ACLOUD were obtained over sea ice in slightly unstable or stable stratification where turbulent heat fluxes are rather small (heat fluxes in the order of a few $\mathrm{W\,m^{-2}}$). Such low flux conditions represent a challenge to instrumentation and measurement strategy and lead to less relative accuracy compared to turbulent fluxes derived in strong convective condition as e.g. cold air outbreaks.*

**p6,l32-34: Run-on sentence, re-write**

We changed this section. See comment above. Parts of these lines where shifted to Section 4.1.2 (new 5.1.2), which reads now as:

*The combination of both aircraft allowed flying vertical stacks at a number of different locations along the mean wind direction. At each stack profiles of mean*

[Figure]

*variables and of turbulent fluxes can be derived. Depending on the structure of the boundary layer, horizontal legs in up to seven altitudes were flown. The typical length of these horizontal sections was at least 10 km, sufficient to apply the eddy covariance method to calculate turbulent fluxes (see Section 3.1). As demonstrated by an example of a single flux profile in Wendisch et al. (2019, Fig. 18), the derived profiles are in agreement with theory showing downward heat fluxes in stable environment and upward fluxes in a well-mixed surface forced convective layer.*

**p7,l11: 12-15 nm: Is this sampling or resolution? If resolution, what is the sampling?**

Yes, this was not clearly defined and is now changed into:

*Two types of grating spectrometers are applied by the SMART-Albedometer. At wavelengths below 920 nm, the spectrometers provide a 1 nm sampling resolution (520 spectral pixel) with a spectral resolution of 2–3 nm full-width of half-maximum (FWHM). Longer wavelengths, 920–2155 nm, 247 spectral pixel, the near-infrared spectrometers sample every 5 nm with a coarser spectral resolution of 12–15 nm*

**p7,l21: remain -> maintain**

Changed as suggested.

**p8,l10: convoluted -> should this be "convolved"?**

Changed as suggested.

**p8,l14: have been –> were (multiple occurrences throughout document)**

Changed as suggested.

**p9,l1-2: check language (punctuation, numerus)...**

Sentences was rephrased.

**p9,l2: Why is the comparison limited to pitch/roll angle < 2?**

Although, the measurements of the CANON fish-eye camera and the spectral imager AISA Eagle/Hawk where corrected for the aircraft attitude, the correction might introduce uncertainties for larger roll and pitch angles, e.g. due to an improper geometrical calibration. To focus on the comparison of the radiometric calibration, we selected only data within the 2° limit. Anyway, during calm remote sensing flight legs at 3.000 m altitude, the aircraft movement mostly did not exceed these values. We added a short justification of the limit in the revised manuscript:

> *To avoid systematic effects due to the attitude correction, the comparison is limited to measurements, where the aircraft did not exceed a horizontal misalignment of more than 2° in roll or pitch angle.*

**p9,l7: "less than 1%" This is a bit unclear. The deviation of SMART from Eagle is 0.02 at about 0.2 in radiance units (Fig 1e). Isn't that 10%?**

Thanks for identifying this typo. We change the number to 10%.

**p9, l33: The KT-19 is not a broadband radiometer - quite the opposite. Why is it in this section?**

That's true although the KT-19 spectral band is covering several μm. As the KT-19 brightness temperatures fit, in our point of view, to the terrestrial pyrgeometer, we kept it in this section and changed the section title to "Broadband solar, terrestrial radiation and surface brightness temperatures".

**p10,l10: off nadir (along track backwards) is a bit contradictory. Is it along track or off-nadir? If off-nadir, why was it mounted in this way (25 deg off)?**

The viewing geometry is both along track and off-nadir. "Off nadir" could be in roll or pitch angle direction. MiRAC is mounted with 0° roll and 25° pitch. 0° roll indicates, that MiRAC is always looking on the flight track, just backward (25° pitch). To avoid confusion, we added the terms "roll" and "pitch" in the description:

*...pointing about 25° off nadir in pitch direction (along track backwards)...*

**p11,l8: "which successively stem from..." This is unclear. How is the frequency related to the "center of the emission line" and (supposedly) sensed atmospheric level. Isn't this rather a matter of atmospheric opacity (regardless of where the emission line center is located), which translates to the location of the weighting function maximum (in remote sensing terms)? I am probably getting this completely wrong based on the text provided.**

Yes, the reviewer is right, the physical background of microwave radiometer measurements has not been well described. We have rewritten this section into:

*Over the open ocean, where the emissivity of the surface is low, this channel can be used to retrieve the liquid water path. The channels around the 183.31 GHz water vapor absorption line can be used to sense atmospheric moisture. The larger the channels are displaced from the absorption line cen-*

*ter, the lower in the atmosphere the emitted radiation originates. The combination of all spectral channels, therefore, provides information of humidity from different layers.*

**p12,section 2.6: should be completely re-written, too many things are unclear. A sunspot is used for measuring solar radiation and solar irradiance? First off, what is the difference - does this refer to sky radiance vs. direct-beam radiance? The use of "solar" for a wavelength range (if that is the intent) is especially confusing here. How about shortwave, visible or near-infrared? Second, why do sunspots (as in the "sunspot cycle" matter here? Or is it literally a "spot of sunlight"? If so, please don't use that term, which is historically reserved for something else. Third, how can a diaphragm do the "focusing"? If anything, it will diffuse radiation, quite the opposite of focusing.**

We are sorry, that our wording did lead to all the confusion. We tried to improve the section as follows. The spectral range of the Sun-photometer is given now right in the beginning. This allows us to keep the common phrase "direct solar irradiance" which is in our view required to specify the source of the radiation. The potentially misleading description of the optics was simplified by using "aparture" as the general function of these components.

> *It operates a filter wheel with 10 selected wavelengths in the spectral range from 367 nm to 1024 nm. To measure the direct solar irradiance, the optics of the SPTA use an aperture with a field of view of 1°.*

**p13,l19: Please describe the "Monte Carlo method" or cite paper.**

Unfortunately, the manuscript describing the Monte Carlo method is still not finally published and can not be cited here. Therefore, we already gave a short general

description of the method in the section of our original manuscript. In our view, this should be sufficient. More details would require a separate long section and is beyond the aim of the data publication. However, we edited the general description of the Monte Carlo method to make it more understandable and to address another reviewer comment below.

*Therefore, the particle number size distribution (PNSD) was obtained in two consecutive steps. First, the CDP-2 raw PNSD was computed by the probe manufacturer software, which applies the first solution of the Mie theory particle size determination. In the second step, raw PNSD has then been corrected using a Monte Carlo inversion method to ensure equiprobable values to all possible solutions of the Mie theory particle size determination. In order to do so, the particle counts ($N_{\mathrm{raw}}$) from one raw size bin were uniformly distributed into a finer binning ($N_{\mathrm{fine}}$) for a more precise particle size determination and a scattering cross section was computed for each $N_{\mathrm{fine}}$. A diameter was then randomly attributed to each counts of $N_{\mathrm{fine}}$ using the different solution given by the Mie theory with equiprobability and these diameters were distributed into the same original size bins ($N_{\mathrm{cor}}$).*

**p15,l9/10: Give some quantitative specifics on which parameters are provided for "sphericity", "shape" and "mesoscopic cystal complexity". Are all of these numbers? How can they be interpreted by the user? For example, does "shape" provide different information compared to "complexity"?**

To address these open question, we added the following short description of the parameters given in the data base:

*The particle shape is given in the form of nine Fourier coefficients $y_k$ ($k = 1...9$)*

*derived from the 2D scattering pattern. Using these coefficients, the particles can be classified as columnar (maxima for $y_2$ or $y_4$), hexagonal (maxima for $y_3$, $y_6$, or $y_9$). In all other cases the particles are classified as irregular. The particle sphericity is given as a binary information, where all particles having sphericity of 1 are classified as spheres. The particle mesoscopic complexity is expressed with a complexity parameter $k_e$ that is an optical parameter varying roughly between 4 to 6. Discussion of the link between the complexity parameter and the actual particle complexity can be found in Schnaiter et al. (2016).*

**p15,section 3.1.4: What is the size range?**

The size range of PHIPS is 20-700 $\mu$m. We added this number to the text.

**p16,figure 3: Does the MC correction just do the Mie correction as described, or does it also entail other standard correction as developed over the decades for single-droplet counters/sizers?**

Yes, the Monte Carlo method just does the Mie correction. The CDP used in ACLOUD is the CDP mark 2. This CDP-2 includes some modifications on the hardware which makes the coincidence corrections unnecessary, even more so given the low particle concentrations encountered in Arctic Clouds. This CPD-2 also has anti-shattering tips which reduce greatly the presence of shattered particle in the sampling volume. Beside, the particle by particle information provides the inter arrival time between particles which identifies the presence of remaining shattered particles. This technique is also used on the CIP and no shattered particles were detected in either CDP or CIP for the ACLOUD data set. Finally, we have used the latest calibration techniques to certify the sampling area and glass beads were used on site several times to ensure consistency and accuracy.

In the revised manuscript we tried to make this more clear by exchanging "CDP" with "CDP-2" and writing:

> *The final calibrated PNSD are obtained by apply the calibrated sampling area and removing shattered particles which are identified from the inter-arrival times.*

**p17,l14 vs. l16: Throughout the manuscript, different values are stated for the upper size limit of the isokinetic inlet and the instruments behind it. While not contradictory, I recommend going through the paper again to make sure there's consistency.**

It's true, that the different upper size limits can lead to confusion. The upper particle size is determined by two things. First the loss in the sampling line and second due to the instrument itself. E.g., the OPC is limited by the sampling line losses, while the CPC upper size is determined by the CPC. In the revised version we tried to make this more obvious. In the instrumentation table we added the upper limit of the OPC.

> *Due to losses in the aerosol inlet and in the CVI sampling lines the upper size limit of the OPC was estimated to about 5 µm.... This way, number concentration of particles down to diameters of 10 nm and up to 3 µm (limited by the CPC) were measured...*

> *Note that these transmission refers only to the main inlet (tip and main 19 mm manifold without additional sampling lines) and not to the individual instruments which have different particle size ranges (see Table 2).*

**p17: Why was no nephelometer flown? Seems standard equipment.**

The main focus of the ACLOUD campaign was to study aerosol-cloud interaction. Characterizing the hygroscopic growth of aerosol particles in this case was of minor importance. Thus, a Nephelometer was not included in the payload. As space, weight, and especially power are limited in polar aircraft, the choice of instruments had to be carefully considered with respect to the main scientific objectives.

**p17: Are all the aerosol size distributions "dry" or measured at ambient humidity (unlikely). Was f(RH) measured (unlikely if no nephelometer was flown). Why is aerosol humidification considered irrelevant for this particular campaign?**

The aerosol size distributions were measured under dry conditions. Due to the temperature increase from outside to inside, a representative measurement of the aerosol hygroscopic growth would not have been possible, because most particulate water evaporates in the sampling lines inside the aircraft cabin. Regarding aerosol particle composition, water evaporates upon entry if the vacuum system of the aerosol mass spectrometer, such that the water content of aerosol particles can not be determined. We did not put more effort into characterizing the aerosol humidification as the main focus of the ACLOUD campaign was to study aerosol-cloud interaction. In the revised manuscript we added a statement, that all aerosol size distributions refer to dry aerosol.

*All aerosol particles sizes measured during ACLOUD refer to dry aerosol, because most particulate water evaporates in the sampling lines connecting the inlets and the instruments due to the higher temperature inside the aircraft cabin.*

**p18: How do the rBC measurements by the SP-2 fit in with the rest of the aerosol measurments?**

The measurement techniques of SP-2 and UHSAS or OPC are different. While

UHSAS and OPC use an optical method to derive the particle size, the diameter of rBC is inferred by the SP-2 from incandescence. Therefore, comparing the SP-2 with UHSAS and OPC does not tell anything relevant on the quality of the data. A combined analysis of the data can only addresses qualitative changes of the particle sizes. That's why we did not include a comparison in the manuscript.

**p19,Table 3: Listing species is helpful, but doesn't per se allow attribution of aerosol type/source. How can the information from the different sensors (PSAP, neph if applicable, SP-2, mass spectrometer) be combined to retrieve broader aerosol typing? In isolation, the information provided here may not be helpful to the data user.**

Here we partly disagree with the reviewer, because we see the data publication in another light. The data paper does not aim to provide the recipes to interpret and analyze the data set. In this data paper we only can describe how the data set is derived and what is included. The analysis/interpretation of the aerosol typing and the combination of different data sets is much more complex and depends on the specific scientific question. This is beyond the aim of this manuscript but will be part of future publications.

**p19,l17: "Therefore" does not mean "Dafuer" Use "to that end" or "to achieve that" or "to do that"**

Changed as suggested. Also in other section of the manuscript.

**p21,l16: "not detectable by the CVI" As written, this sentence suggests that under different circumstances (higher ice crystal concentrations), the CVI would be able to distinguish between crystals and droplets. But is that true?**

We do not directly claim this, but in theory it would be possible. When the ice crystal concentrations are higher and the size of the droplets is smaller than the ice crystals, which is likely due to the Wegner-Findeisen-Bergeron process, a high CVI cut-off would solely select the ice crystals. However, during ACLOUD few large droplets were always present in higher concentration than the ice particles in the same size range. Therefore, it was not possible to collect only ice particles in the clouds. Due to the dominance of the liquid droplets, we state, that the majority of cloud particle residuals sampled during ACLOUD can be considered to represent cloud droplet residuals.

**p21,l26: "with respect to" -> "relative to"?**

Changed as suggested.

**p21,l31: This begs the question what the detection limit of the instruments behind the counterflow impactor (or regular inlet) are...**

Yes, this sentence was imprecise. The detection limit, if interpreted as particle cut off, is not improved for all instruments. It is rather the number of particles counted by the instruments, which increases. Therefore, we changed the sentence.

*This has the positive effect to counting statistics of the connected instruments.*

**p22,l1: "extend" -> "extent"**

Changed as suggested.

**p22,l26: How does the transmission (=sampling efficiency?) fall off after 1 micron?**

Calculations give a particle transmission of 80% at 5 μm and to 30% at 10 μm. We added this information in the revised manuscript.

> *Sampling speed in the inlet tip was approximately isokinetic for the airspeeds during ALOUD, such that the particle transmission by the inlet was near unity for particles from 20 nm to about 1 μm and falls to 80% at 5 μm and to 30% at 10 μm. Note that these transmission refers only to the main inlet (tip and main 19 mm manifold without additional sampling lines) and not to the individual instruments which have different particle size ranges (see Table 2).*

**p24,l7: "loosing data" -> "losing data"**

Changed as suggested.

**p24,l1: "aims to characterize" -> "aims at..."**

Changed as suggested. Also in other section of the manuscript.

**p24,l9: "exemplary" does not mean "beispielsweise" - it means "outstandingly good" (or sometimes "serving as a deterrent/bad example") - please revise unless you mean one of the two.**

"exemplary" was removed

**p24,l25: "rather" -> "more or less" ?**

Changed as suggested.

**p25,l2: "out-side" -> "outside"**

Changed as suggested.

**p25, section 4: This would be the opportunity to give some examples how the P5/P6 together achieve the general goals of the campaign, but this is only done in terms of instrument synergies, and not in terms of the fulfillment of mission requirements/goals. Of course, that would only be possible if those were stated at the beginning. I strongly recommend that the authors consider adding such a description as proposed in the general comments. The manuscript would benefit tremendously form that addition (IMO).**

In the revised manuscript, we added a more specific description of the research flights and refer to the campaign overview paper (Wendisch et al., 2019), where all scientific objectives of the mission are explained. If these scientific objectives are fulfilled, we can only answer, when the data is fully analyzed. This is in our view not the focus of the data paper. Here we only want to demonstrate, that the data acquisition, flight strategy, and combination of data from both aircraft was successful. We hope, that the manuscript clearly demonstrates, that the acquisition of the data set was successful. Everything beyond needs to be addressed by scientific studies using the data.

**p27,Figure 8: There are not enough details provided. How about adding imagery for context, and a cross section of one of the short "fishbone" segments to allow the reader to see the vertical structure? How does the length of the short/long legs optimize the sampling / how does it fulfill mission requirements? This is very useful information for the reader.**

Imagery of the individual flights will be provided in the supplementary material. A vertical cross section was added to the figure. To describe, how these flight pattern and

the combined flight contributed an improved data analysis, we rewrote this subsection as:

> *The combination of both aircraft allowed for flying vertical stacks at a number of different locations along the mean wind direction. At each stack profiles of mean variables and of turbulent fluxes can be derived. Depending on the structure of the boundary layer, horizontal legs in up to seven altitudes were flown. The typical length of these horizontal sections was at least 10 km, sufficient to apply the eddy covariance method to calculate turbulent fluxes (see Section 3.1). As demonstrated by an example of a single flux profile in Wendisch et al. (2019, Fig. 18), the derived profiles are in agreement with theory showing downward heat fluxes in stable environment and upward fluxes in a well-mixed surface forced convective layer. To study the change of flux profiles along the mean flow, series of vertical stacks were flown on 14, 20, and 25 June 2017 (Flights #17, #21, #23). As an example the flight track of 25 June 2017 is illustrated in Figure 8. Compared to a single aircraft mission, the number of locations available for analyzing flux profiles was increased by a factor of two without reducing the length of the horizontal legs. The six locations of the vertical stacks (six legs with each 30 km length) extend over a distance of 170 km with 28 km horizontal separation of the individual profiles. However, for the combined analysis, it needs to be assured that measurements of instruments on both aircraft can be merged into a single data set.*